# Unlocking the power of L1 regularization: A novel approach to taming overfitting in CNN for image classification

Ramla Sheikh[1], Fazli Wahid[1,2,3], Sikandar Ali[1], Ahmed Alkhayyat[4,5,6], Yingling Ma[3], Jawad Khan[7]*, Youngmoon Lee[8]*

**1** Department of Information Technology, The University of Haripur, Haripur, Pakistan, **2** Collage of Science and Engineering, School of Computing, University of Derby, Derby, United Kingdom, **3** School of Computing Sciences, University of East Anglia, Norwich, United Kingdom, **4** College of Technical Engineering, The Islamic University, Najaf, Iraq, **5** Department of Computers Techniques Engineering, College of Technical Engineering, The Islamic University of Al Diwaniyah, Al Diwaniyah, Iraq, **6** Department of Computers Techniques Engineering, College of Technical Engineering, The Islamic University of Babylon, Babylon, Iraq, **7** School of Computing, Gachon University, Seongnam, Republic of Korea, **8** Department of Robotics, Hanyang University, Ansan, Republic of Korea

* jkhanbk1@gachon.ac.kr (JK); youngmoonlee@hanyang.ac.kr (YL)

## Abstract

Convolutional Neural Networks (CNNs) stand as indispensable tools in deep learning, capable of autonomously extracting crucial features from diverse data types. However, the intricacies of CNN architectures can present challenges such as overfitting and underfitting, necessitating thoughtful strategies to optimize their performance. In this work, these issues have been resolved by introducing L1 regularization in the basic architecture of CNN when it is applied for image classification. The proposed model has been applied to three different datasets. It has been observed that incorporating L1 regularization with different coefficient values has distinct effects on the working mechanism of CNN architecture resulting in improving its performance. In MNIST digit classification, L1 regularization (coefficient: 0.01) simplifies feature representation and prevents overfitting, leading to enhanced accuracy. In the Mango Tree Leaves dataset, dual L1 regularization (coefficient: 0.001 for convolutional and 0.01 for dense layers) improves model interpretability and generalization, facilitating effective leaf classification. Additionally, for hand-drawn sketches like those in the Quick, Draw! Dataset, L1 regularization (coefficient: 0.001) refines feature representation, resulting in improved recognition accuracy and generalization across diverse sketch categories. These findings underscore the significance of regularization techniques like L1 regularization in fine-tuning CNNs, optimizing their performance, and ensuring their adaptability to new data while maintaining high accuracy. Such strategies play a pivotal role in advancing the utility of CNNs across various domains, further solidifying their position as a cornerstone of deep learning.

**Data availability statement:** We used the following benchmark datasets, which are freely available online: • MNIST: Available at https://www.kaggle.com/datasets/oddrationale/mnist-in-csv • Mango Trees Leaf: Available at https://data.mendeley.com/datasets/94jf97jzc8/1 • Hand Drawn Sketches Images (Tree Category): Available at http://cybertron.cg.tuberlin.de/eitz/projects/classifysketch/ Paper code for replication will be available at https://github.com/hqsikandar/L1-REGULARIZATION-for-CNN.

**Funding:** This work was supported by Institute of Information and Communications Technology Planning and Evaluation (IITP) grant IITP-2025-RS-2020-II201741, RS-2022-00155885, RS-2024-00423071 funded by the Korea government (MSIT).

**Competing interests:** The authors have declared that no competing interests exist.

## Introduction

Deep learning, a subset of artificial intelligence and machine learning, is a computational framework that facilitates systems to learn step-by-step representations of data through continuous training. It has achieved remarkable success in natural language processing (NLP), Image classification, and speech recognition. Deep learning allows machines to acquire knowledge from experience without explicit human invention. Since it has begun by Hinton et al. In 2006, Deep Learning revolutionized artificial intelligence, especially in image classification. Convolutional Neural Network (CNN) has been given state performance by taking advantage of hierarchical traction and spatial invariance. Recent progress, such as skilled network designs, transformer-based architecture, and mobile video networks (films) have expanded deep learning skills in handling complex visual data. Its popularity arises from its ability to achieve exceptional accuracy and outperform other network architectures when properly trained. Deep learning analyzes the vast amount of unstructured data by processing numerous features. The deep learning algorithm consists of multiple layers, each designed to extract and examine distinct features from the data. The input layer extracts features at appropriate levels and passes them to subsequent layers through iterations. While the initial layer captures basic information, the deeper layers build on this to create more comprehensive and abstract representations [1].

### Deep learning approaches for image classification

Mobile Video Networks (MoViNets) are designed for efficient video understanding and image classification on mobile devices. MoViNets optimize computational efficiency while maintaining high accuracy [2]. Twins introduced a dual-stream architecture that uses transformer encoders for visual and textual information, achieving strong performance in vision-language tasks such as image captioning and visual question answering [3]. Patch Pairwise Vision Transformer with Attentive Spatial Embeddings (PPV-ASE) leverages patch-wise pairwise relationships and spatial embedding. PPV-ASE enhances image classification performance, particularly in tasks requiring fine-grained feature extraction [4]. Cross-modality training (CMT) is a multi-modal model that combines image and text data to improve image classification and retrieval tasks, enabling models to leverage complementary information from **data** [5]. RegNet focuses on designing scalable network architectures and achieves strong performance in image classification tasks, making it suitable for both small and large datasets [6]. Lambda Network integrates global context information through lambda layers, enhancing image recognition capabilities [7].

| Architecture | Medical Application | Key Advantage |
|---|---|---|
| MoViNets | Mobile diagnostics | Real-time processing |
| PPV-ASE | Histopathology | Fine-grained feature extraction |
| CMT | Radiology reports | Multimodal fusion |
| L1-CNN | All domains | Sparse, interpretable models |

## Convolutional neural networks (CNN)

CNNs are a pivotal algorithm in deep learning, particularly for image classification and analysis. They apply neural networks to two-dimensional arrays, like images, using localized neural input, shared weights, and spatial down-sampling. CNNs employ convolution operations, which allocate weights and biases to image components, making them highly efficient for image processing and requiring less preprocessing than other methods [8]. Prominent CNN architectures include VGGNet, GoogLeNet, LeNet, ResNet, AlexNet, and ZDNet, serving various image-related tasks. CNNs rely on essential components, such as convolutional layers with filters, pooling techniques, appropriate activation functions (e.g., rectified linear units), and loss functions. For model training. To combat overfitting, regularization techniques like L1 regularization, L2 regularization, and dropout are used, enhancing the robustness and generalization of CNN models [9].

## L1 regularization and its role in overcoming overfitting

The The methods adapted for handling general unconstrained differentiable loss functions primarily emphasize a single scalar parameter, λ. However, it is worth noting that these techniques can be readily extended to accommodate a separate λ value for each element. These additional λ values can be set to zero when necessary to avoid penalizing specific elements. Unless specified otherwise, the stability of the algorithms is ensured to guarantee global convergence. This is achieved through the implementation of a backtracking line search that identifies an appropriate step length, denoted as "t," in accordance with the Armijo condition. To generate trial points during this process, we employ cubic interpolation techniques that take into account both function values and directional derivative values. Furthermore, a sufficient decrease parameter of 0.0001 is applied to validate the chosen step length [10]. To create an efficient sparse convolutional neural network and combat weight redundancy arising from matrix multiplication, we utilize L1 regularization during optimization. This regularization method, ideal for scenarios involving many features, promotes sparsity and computational efficiency, aiding in feature selection. It is applied to various components of the dense layer, including kernel weights, biases, and activity. Weight regularization is added to the dense layers to reduce overfitting. The loss function comprises an error term and an L1 penalty term, with a tuning parameter (λ) controlling the regularization strength—ensuring both error minimization and weight shrinkage in the model [11].

One approach is the autoencoder scheme, which involves a neural network that compresses and decompresses data to eliminate noise. Stacked autoencoders are used. Data augmentation expands the training dataset with transformations to reduce overfitting, including Gaussian noise control. Batch normalization addresses internal covariate shift and speeds up training. L1 regularization (LASSO) removes irrelevant features by penalizing weights based on their absolute values. These techniques collectively contribute to regularization and noise reduction in the neural network model [12]. To diversify Pareto solutions, maintain $\alpha_n$ at L1 while using a specific method to set the search direction vector λ, thus striking a balance between loss and L1-Regularization weight. Additionally, an adjustment is applied to evaluation values to balance the influence of objective functions by using logarithmic loss and the second objective function's mean values across the initial population. This helps select individuals with smaller L1 norms as learning progresses. In the Focused transformation approach, a weighted sum function is employed for scalar fitness, considering both the loss and L1 norm, ensuring a more comprehensive optimization strategy [13].

When applying L1 regularization to a CNN, it involves introducing a penalty term with a specified coefficient value to the network's dense (fully connected) layers. This regularization process encourages many of the weight values in these dense layers to become small or even zero. This effect simplifies the model's capacity to capture and represent features in the data. By selectively attenuating certain connections, L1 regularization helps the CNN identify and emphasize the most relevant features while reducing the impact of less important ones.

In the context of CNNs, applying L1 regularization can be beneficial for preventing overfitting, promoting sparsity in learned weights, and leading to a more compact and interpretable representation of the data. The regularization enhances

the model's generalization performance and its ability to make accurate predictions by focusing on essential features while reducing noise or unnecessary complexity in the network. This approach can improve the model's efficiency and effectiveness in various classification or analysis tasks.

## Literature review

### Standard convolutional neural network

Numerous authors in the literature have employed standard Convolutional Neural Networks (CNNs) for various image classification tasks. For instance, Jiaji Wang et al [14] examine Convolutional Neural Networks (CNNs) and their applications in medical image processing, with an emphasis on design improvements, overfitting prevention approaches, and the usage of pre-trained models to get better outcomes. It teaches how CNNs function, covering layers for image processing, data reduction, and decision-making, as well as how to deal with difficulties such as noise-induced mistakes. The research demonstrates how pre-trained models (such as AlexNet and ResNet) may assist assess tiny medical datasets. It also discusses how CNNs are used to diagnose disorders in the brain, heart, lungs, and breasts utilizing techniques like MRI, CT scans, and X-rays. The objective is to develop dependable, easy-to-comprehend, and efficient AI systems to improve healthcare diagnosis and treatment.

M. Agarwal et al [15] focused on detecting and categorizing diseases affecting tomato crops using a deep learning-based approach. Their model incorporated three convolutional layers, three max-pooling layers, and two fully connected layers, outperforming pre-trained models like VGG16, InceptionV3, and MobileNet with an accuracy of 77% for disease classification. Similarly, Justice O. Emuoyibofarhe et al [16] compared three different CNN models trained on skin images, achieving a 90% training and 81% testing accuracy with Google Inception V3. Meanwhile, Rohit, Akshit, et al [17] utilized CNN with the MNIST dataset, achieving 70% accuracy for certain digits and 77% for others.

K. Kusrini et al [18] employed a pre-trained VGG-16 model with a 2-layer fully connected network, achieving accuracies ranging from 67% to 75% in different versions of their model. L. Zhang et al [19] achieved a 75% classification accuracy using a CNN. R. Sharma et al [20] applied CNN to identify and forecast illnesses in rice crops, potentially saving yields from substantial losses. Hao Wu and Zhi Zhou [21] developed a DL-based AI system with 91% accuracy in distinguishing between normal and faulty images

P. Lakshmi Prasanna et al [22] implemented image categorization with CNN, achieving 90.32% accuracy on the test set and 93.58% on the training set using a hierarchical model. Alshazly H et al [23] introduced CovidDenseNet and CovidResNet models for COVID-19 detection, reaching up to 93.87% accuracy in binary classification. Other authors have also employed standard CNNs in various image-classification contexts. Wei Fang et al [24] improved CNN-based image recognition, and Yunendah Nur Fuadah et al [25] used CNNs to automatically identify benign tumor lesions and skin cancer, with the Adam optimizer performing optimally for classifying skin lesions with the ISIC dataset.

### Modified convolutional neural network

To enhance CNN's performance, researchers have introduced modifications and innovations to its core mechanisms. For instance, S. Kausar et al [26] utilized CNN to predict the total number of teachers in Pakistani educational institutions, demonstrating the potential for implementing new teacher policies based on their model's 89.485% accuracy. M. H. Masood et al [27] proposed a novel approach for localized categorization of diseased sections within images, achieving an overall accuracy of 87.6% for assessing agricultural damage. In another study, J. Velasco et al [28] employed the MobileNet model for classifying skin illnesses, exploring different sampling strategies and preprocessing techniques to achieve accuracies ranging from 84.28% to 93.6%.

Saravanan Srinivasan et al [29] offers three alternative CNN models made for various categorization tasks to improve early detection using a deep convolutional neural network (CNN). The first CNN model detects brain cancers with an

astounding 99.53% accuracy rate. The second CNN model effectively classifies brain cancers into five different types: normal, glioma, meningioma, pituitary, and metastatic, with an accuracy of 93.81%. Additionally, the third CNN model classifies brain tumors into their various classes with an accuracy of 98.56%. A grid search optimization technique is used to automatically adjust all pertinent CNN model hyperparameters in order to guarantee peak performance. Strong and trustworthy classification findings are obtained by using sizable, openly available clinical datasets.

A. Hussain et al [30] used CNN to classify wheat diseases, achieving an accuracy of 84.54%, offering a valuable tool for farmers to protect their wheat crops. S. Ghosal et al [31] tackled rice leaf blight using a VGG-16-based CNN architecture, achieving a 97% training accuracy and a 92.4% testing accuracy. Ul Khairi et al [32] tackled fine-grained vehicle categorization challenges using multiple datasets and DCNN models, achieving classification accuracies ranging from 78% to 87%.

Similar to these studies, other researchers have employed upgraded CNN models for various image categorization tasks. For example, Kang IL Bae et al [33] introduced a modified m-CNN strategy for multimodal categorization, while Zhiguan Huang et al [34] proposed CNNBCN for brain cancer classification. Haidong Shao et al [35] developed a CNN framework for rotor-bearing system failure diagnosis, and Guangyu Jia et al [36] focused on COVID-19 diagnosis using CXR and CT images. Yi Wang et al. [37] created a CNN-based system for breast lesion diagnosis, and Lima Hussain et al [38] compared different CNN architectures for cervical lesion detection.

## Hybrid convolutional neural network

To boost CNN's capabilities for image classification, researchers have explored hybrid approaches that combine CNN with other machine learning or deep learning models. For instance, Oluwaseun Ajao et al [39] introduced a hybrid CNN and LSTM model for fake news identification, achieving improved prediction accuracy by incorporating both text and image features.

Savita Ahlawata and Amit Choudhary [40] proposed a hybrid CNN-SVM model for automatic feature generation. In this model, SVM replaces the Softmax layer of CNN and operates as a binary classifier. This approach achieved an impressive 99.28% recognition accuracy on digit handwritten images. Osman Doğuş Gülgün and Hamza Erol [41] presented hybrid CNN models for medical image classification. Their models extracted features from various medical images, including brain MRIs and lung x-rays, achieving high accuracy for tumor detection and pneumonia classification.

Ashutosh Kumar Singh et al [42] employed data augmentation and various deep learning techniques, including CNN, to enhance crop quality and identify plant diseases, achieving promising results in detecting illnesses in various plants. M. Ahmad et al [43] compared different techniques, including SVM and CNN, for disease detection. They found that CNN achieved superior accuracy levels, especially when combined with data augmentation and a triple dataset.

M. M. Srikantamurthy et al [44] Using the BreakHis dataset, the team created a hybrid CNN-LSTM model to categorize four kinds of breast cancer: benign and malignant. With 99% accuracy for binary classification (benign vs. malignant) and 92.5% accuracy for multi-class classification of subtypes, the model, which included transfer learning, beat other models such as VGG-16 and ResNet50. The optimizer with the highest accuracy and the lowest loss was the Adam optimizer. There is a great chance that this hybrid technique will accurately classify breast cancer.

Deshpande UU et al [45] introduced a minutia-based CNN matching model for fingerprint identification, achieving identification rates of 80% and 84.5% on the FVC2004 and NIST SD27 datasets. M. U. Rehman et al. [46] proposed a deep learning architecture combining 3D CNN and LSTM for video-based classification, reaching an impressive 97% accuracy on their dataset.

Karungaru Stephen et al [47] improved AlexNet for vehicle detection and classification, achieving faster classification speeds and better generalization using hybrid CNN-SVM models. Xuping Gong and Yuting Xiao [48] used CNN and NLP technology to create an interactive skin cancer detection website, improving accuracy through CNN parameter adjustments. Rajmodhan et al [49] utilized a hybrid CNN and SVM model for smart paddy crop disease detection. The important thing,

according to Mohamed et al. [50], is to smooth the standard regularization term at the origin. In addition to producing sparse and effective neural networks, this processing offers a theoretical understanding of the algorithm. Second, to increase the network learning speed even further, add the adaptive momentum term to the iteration process. Furthermore, numerical studies demonstrate that the suggested technique boosts the computation learning rate and removes oscillation.

These studies demonstrate the effectiveness of hybrid models in various domains, leveraging CNN's strengths in feature extraction and classification while incorporating additional techniques to enhance performance.

Recent studies determine the efficiency of CNNs in image classification in medical, agricultural, and industrial fields, with standard models achieving 70–90% accuracy but having overfitting. Modified architectures (VGG-16, Mask R-CNN) improve performance to 84–95% accuracy, while hybrid approaches (CNN-SVM, CNN-LSTM) reach up to 95% accuracy in medical imaging, whereas they require more computational resources. Key encounters include data dependency, generalization gaps, and high computational costs, with medical applications outperforming agricultural applications due to standardized datasets.

**Summary Table:**

| S. No | Author | Methodology | Accuracy | Dataset | Limitation |
|---|---|---|---|---|---|
| 1 | Rohit, Akshit et al. | Combined CNN with MNIST dataset | 70% (some digits), 77% (others) | MNIST dataset | Inconsistent accuracy due to overfitting |
| 2 | K. Kusrini et al. | Pre-trained VGG-16 + 2-layer FC network | Version 0: 70% (train), 67% (test) Version 1: 75%, 68% Version 2: 71%, 74% | Mango Dataset | Lower test accuracy suggests overfitting |
| 3 | L. Zhang et al. | CNN for classification | 75% | Hand-drawn sketches | Limited generalization on test data |
| 4 | R. Sharma et al. | CNN for rice crop disease detection | 90.32% (test), 93.58% (train) | Rice Crop | Slight overfitting observed |
| 5 | Justice O. Emuoyibofarhe et al. | Compared 3 CNNs for skin cancer classification | 81% (testing) | Skin cancer images | Overfitting due to small dataset |
| 6 | Zarrim Tasmin et al. | CNN for colon cancer detection | 95%–99% | Colon cancer images | Performance variability due to overfitting |
| 7 | Alshazly H et al. | Proposed CovidDenseNet & CovidResNet | 81.77% | SARS-CoV-2 CT scans | Overfitting in binary classification |
| 8 | S. Ghosal et al. | VGG-16-based CNN for rice leaf blight | 97% (train), 92.4% (test) | Rice crop | Generalization gap indicates overfitting |
| 9 | S. Kausar et al. | CNN for teacher workforce prediction | 89.485% | Teacher hiring data | Potential overfitting on training data |
| 10 | M. H. Masood et al. | Mask R-CNN for localized disease patches | 87.6% | Plant disease dataset | High computational complexity |
| 11 | A. Hussain et al. | CNN for wheat disease classification | 84.54% | Wheat crop dataset | Overfitting due to limited samples |
| 12 | J. Dong et al. | Modified CNN for skin cancer classification | 89.5% | Skin cancer dataset | Overfitting observed in training |
| 13 | Oluwaseun Ajao et al. | Hybrid CNN-LSTM for fake news detection | 74% | Twitter posts | Complex architecture leads to overfitting |
| 14 | Osman Dogus Gulgun et al. | Hybrid CNN-SVM for medical image classification | 85%–92% | Brain MRI & lung X-ray images | Accuracy fluctuations due to overfitting |
| 15 | Ashutosh Kumar et al. | CNN-SVM for plant disease detection | 96.1% | Plant dataset | Possible underfitting in some classes |
| 16 | Deshpande UU et al. | Minutiae-based CNN for fingerprint identification | 84.5% | FVC2004 & NIST SD27 datasets | Overfitting due to high model complexity |
| 17 | Xuping Gong & Yuting Xiao | CNN-NLP hybrid for skin cancer detection | 83% | Skin cancer dataset | Overfitting in deep feature extraction |

## Methodology

The proposed methodology involves training a Convolutional Neural Network (CNN) while strategically applying L1 regularization to different layers of the model with varying L1 values on a specific dataset. In this approach, the CNN's architecture, including the number of convolutional and pooling layers, filter sizes, and the number of neurons in the fully connected layers, is designed to shape the model's capacity and complexity. The key parameter here is the L1 regularization strength (lambda), which determines the extent of the penalty applied to the model's weights. The design of CNN, including layers and filters, is carefully established how complicated the model is. L1 regularization provides a penalty term for loss function, calculated as in equation 1:

$$Loss = OriginalLoss + \lambda \sum i \vee wi \vee \tag{1}$$

Here, $wi$ represents the weight of the model. A higher λ value simplifies the model by reducing unnecessary weight, while a lower λ value applies less regularization. In this method, higher λ values are used in dense layers to help choose important features and prevent overfitting, while lower λ values are spent on convolutional layers to hold important spatial details. For example:

1. On the MNIST dataset (handwritten numbers), the λ value of the 0.01 model simplifies, which improves its capacity to generalize.

2. On the mango tree, using two λ values (e.g., 0.001 for the convolutional layer and 0.01 for dense layers), makes the model better in classifying the leaves.

3. On the Quick, Draw! dataset (diverse sketch), λ value of 0.001 helps the model identify different sketches more accurately.

This adaptive L1 regularization model balances the model complexity and Feature preservation, making the CNN robust, easy to understand, and successful to handle different tasks. By carefully adjusting λ values for each layer, the method ensures that the model performs well and avoids overfitting, making it a useful tool to improve the CNN performance, as shown in Fig 1.

## Proposed algorithm

**Pseudo code 1**

```
1) Define the architecture of the convolutional neural network, including the convolutional layers,
   pooling layers, dense layers, and activation functions.
a. Purpose: To define the structure of the CNN and how data flows across the network.
b. Functionality:
i. Convolutional layers use filters (kernels) to extract features from input images.
ii. Pooling Layers: Reduce the spatial dimensions of the feature maps, increasing model efficiency.
iii.  Dense Layers: Use features from previous layers to create predictions.
iv. Activation Functions: Non-linearity (such as ReLU) is used to assist the model in understanding
    complex patterns.
      Relevance: The architecture defines the model's ability to learn and generalize from data.
2) Define the loss function, which should include both the categorical cross-entropy loss and the L1
   regularization term.
a) Purpose: To measure how properly the model is performing and guide its gaining knowledge of the
   process.
b) Functionality:
a. Categorical Cross-Entropy Loss: Measures the difference among predicted and actual class proba-
   bilities (used for multi-class category).
```

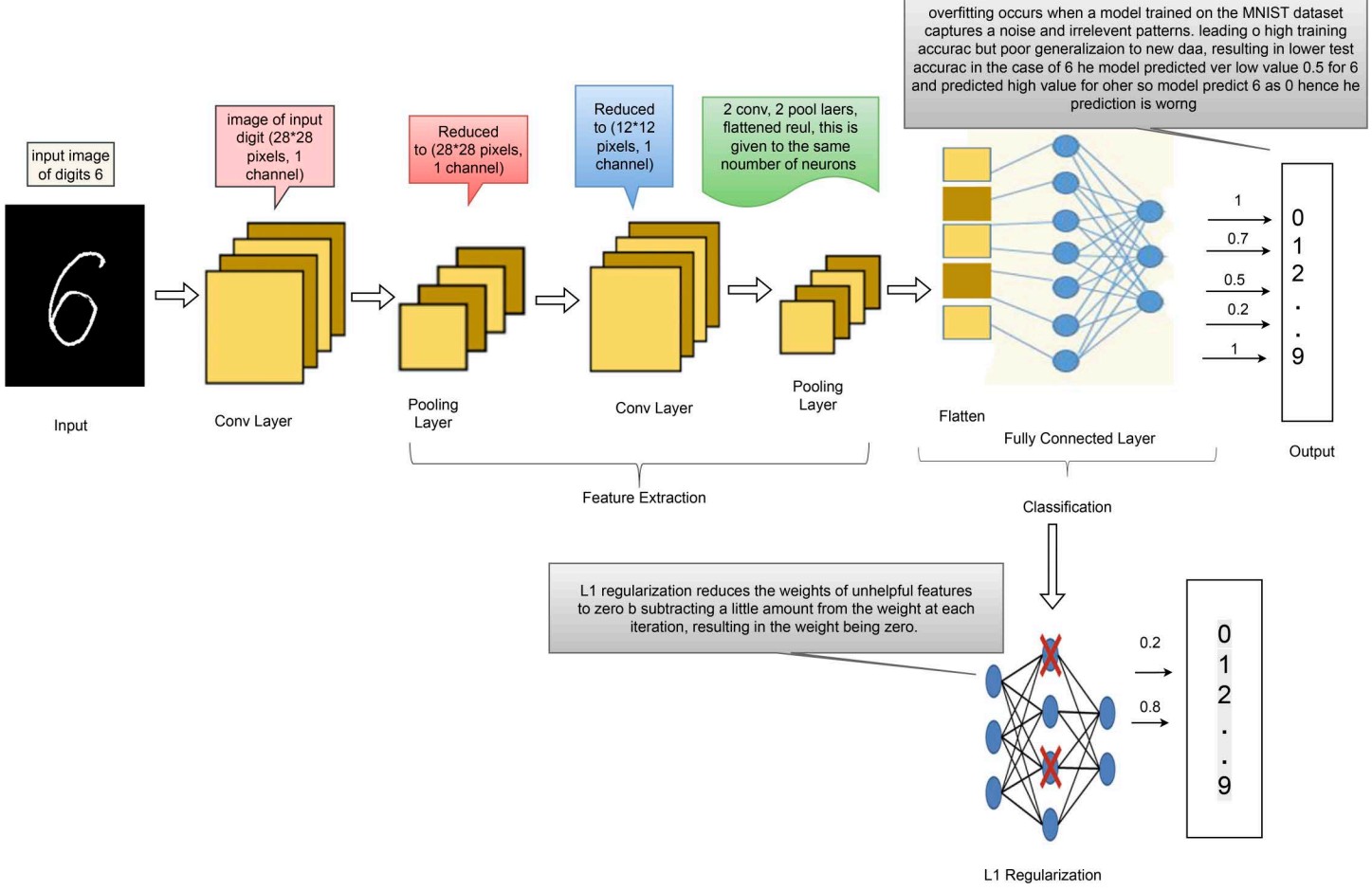

**Fig 1. Proposed model.**

b. L1 Regularization Term: Adds a penalty proportional to the absolute value of the model's weights to inspire sparsity.

c. The total loss is:

d. $TotalLoss = Cross - EntropyLoss + \lambda i \sum \vee wi \vee$

Where $\lambda$ is the regularization strength and $wi$ are the model's weights.

c) Relevance: The loss characteristic ensures the model learns successfully at the same time as avoiding overfitting through regularization.

3) Initialize the weights and biases of the model.

a. Purpose: To set the initial values of the model's parameters before training.

b. Functionality:

i. Weights and biases are initialized randomly or the use of precise strategies to ensure the model learning effectively.

c. Relevance: Proper initialization enables the model converge quicker and avoid issues like vanishing or exploding gradients.

4) Iterate over the training data, using each sample to predict with the current model parameters.

a) Purpose: Training the model using the available data.

b) Functionality:

i. For each image in the training dataset:

ii. Pass the image through the CNN to generate results.
   Relevance: Step 4 allows the model to learn patterns from the data
5) Calculate the loss for the sample by comparing the predicted output to the actual output.
a) Purpose: To calculate how far the model's predictions are from the actual labels.
b) Functionality:
a. Compare the predicted output (from Step 4) with the actual output (ground truth) using the loss
   function defined in Step 2.
c) Relevance: The loss calculates the model's performance and leads to parameter updates
6) Calculate the gradients of the loss with respect to the model parameters.
a) Purpose: To determine how changes in the model's parameters affect the loss.
b) Functionality:
a. Use backpropagation to compute the gradients of the loss concerning each weight and bias in the
   model.
b. Gradients specify the direction and size of updates needed to minimize the loss.
c) Relevance: Gradients are important for updating the model's parameters efficiently
7) Update the model parameters by subtracting the gradients multiplied by the learning rate.
a) Purpose: To develop the model's performance by correcting its weights and biases.
b) Functionality:
a. Update each parameter (weight or bias) using the formula:

$$wi = wi - (1 + \eta x)^n + \frac{n \partial Loss}{2! \partial wi} + \ldots$$

   Where η is the learning rate (controls the size of updates).
c) Relevance: Step 7 makes sure the model learns from its mistakes and improves over time.

   Evaluate the performance of the model on a validation set or test set to assess its generalization performance. This pseudo code defines a simple convolutional neural network with L1 regularization architecture with multiple convolutional and pooling layers and a dense layer with a softmax activation function. The loss function combines the categorical cross-entropy loss and the L1 regularization term. The optimizer updates the parameters using gradient descent. The learning rate and regularization strength can be set as needed. This is a general outline of the procedure, and the details of the implementation will depend on the specific problem and requirements of the model Pseudo-code for implementing L1 regularization in a convolutional neural network:

## Pseudo code 2
**Pseudocode of Convolutional neural network with L1 regularization**
```
1.  # Define the model architecture
2.  function model(X, W1, b1, W2, b2,...,
    Wn, bn)
3.  conv_layer1=convolutional layer with parameters W1 and b1
4.  pool_layer1=pooling layer
5.  conv_layern=convolutional layer
    with parameters Wn and bn
6.  pool_layern=pooling layer
7.  flatten=flatten layer
8.  dense=dense layer with parameters W
    and b
9.  output=softmax activation
10. return output
11. end
12. # Define the loss function
13. function loss_fn(y, y_pred, W1, W2,..., Wn, lambd)
14. categorical_crossentropy =
    cross-entropy loss of y and y_pred
15. l1_reg=sum of absolute values of W1, W2,..., Wn multiplied by lambd
```

```
16. return categorical_crossentropy+l1_reg
17. end
18. # Define the optimizer
19. function train_step(X, y, W1, b1,
    W2, b2,..., Wn, bn,
    learning_rate, lambd)
20. y_pred=model(X, W1, b1, W2, b2,...,
    Wn, bn)
21. loss=loss_fn(y, y_pred, W1, W2,..., Wn, lambd)
22. dW1=derivative of loss with respect to W1
23. db1=derivative of loss with respect to b1
24. dWn=derivative of loss with respect to Wn
25. dbn=derivative of loss with respect to bn
26. W1=W1 - learning_rate * dW1
27. b1=b1 - learning_rate * db1
28. Wn=Wn - learning_rate * dWn
29. bn=bn - learning_rate * dbn
30. return loss, W1, b1, W2, b2,..., Wn, bn
31. end
```

## Experimental setup and results

### System specification

The hardware used in the suggested methodology is 12 GB RAM, 250 M2 SSD, 500 GB Hard Disk, and Windows 11 64-bit operating system. Convolutional neural network simulation with L1 regularization is done in Python. The code is executed using Jupyter Notebook.

### Data division

The three datasets are used for the implementation of convolutional neural network with L1 regularization The dataset are taken from these websites. The MNIST dataset can be downloaded from this link
https://www.kaggle.com/datasets/oddrationale/mnist-in-csv, and the second dataset mango tress leaf can be downloaded from this link
https://data.mendeley.com/datasets/94jf97jzc8/1, and the tree dataset is hand drawn sketches images dataset can be downloaded from this link http://cybertron.cg.tuberlin.de/eitz/projects/classifysketch/
   Cross-validation is performed on the following ratios:

| Split | Pros | Cons | Best for |
|---|---|---|---|
| 50−50 | Test Extreme data insufficiency | High variance in validation | Small datasets |
| 60−40 | More training data than 50−50 | Still limited for deep learning | Medium datasets |
| 70−30 | Balanced moderate datasets | Validation set may be noisy | Can be common use |
| 80−20 | Best for large datasets | Slightly less validation data | Most CNN applications |
| 90−10 | Maximizes training data | set if validation is too small (risk of overfitting) | Too much data |

1. 70−30% (The training data 70% of the total data. The remaining 30% is set aside for testing.)

2. 60−40% (The training data 60% of the total data. The remaining 40% is set aside for testing.)

3. 50−50% (The training data 50% of the whole dataset. The remaining 50% is set aside for testing.)

4. 80−20% (The training data 80% of the total dataset. The remaining 20% is set aside for testing.)

5. 90−10% (The training data for 90% of the total dataset. The remaining 10% is set aside for testing.)

## Results and discussions

The outcomes of our suggested models are briefly explored in this section. The following are the results of all datasets based on a convolutional neural network with augmenting L1 regularization model.

### MNIST dataset

The experiment is performed for a convolutional neural network with an L1 regularization MNIST dataset. When applying L1 regularization with a coefficient value of 0.01 to the dense (fully connected) layers of a Convolutional Neural Network (CNN) trained on the MNIST dataset, the model undergoes a regularization process where the penalty term encourages many of the weight values in these dense layers to become small or even exactly zero. This effect simplifies the model's capacity to capture and represent features in the dataset. By selectively attenuating certain connections, L1 regularization helps the CNN identify and emphasize the most relevant features while reducing the impact of less important ones. In the context of MNIST, which consists of hand-written digits, applying L1 regularization with a coefficient of 0.01 can prevent overfitting, promote sparsity in the network's learned weights, and lead to a more compact and interpretable representation of the digit images. This regularization can enhance the model's generalization performance and its ability to classify digits accurately. The total training time for the model on the MNIST dataset was approximately 14.9 hours (53,587 seconds), based on 41 training runs. This highlights the computational cost of training a convolutional neural network on a large dataset like MNIST. Furthermore, the average testing time per sample was approximately 5.8 milliseconds (0.0058 seconds), demonstrating the model's efficiency in making predictions on new data. Other factors also have significant effects on the model's performance, speed, and time (e.g., GPU/CPU usage).

The experiment is performed for an 80−20% ratio. The graph shows the fluctuation of training and validation accuracy of the convolutional neural network and L1 regularization model is 99%, shown in Fig 2 convolutional neural network and L1 regularization model training accuracy is 97%, and model validation accuracy is 99.2%. The Y-axis represents training and validation accuracy, while the X-axis represents epoch count. It took 60 epochs for the model to converge on an optimal convolutional neural network and an L1 regularization model for digit recognition. When we first started running our model, it gave less accuracy in training and validation, but as the epoch size increased over time, the results improved and improved until the model gave 97% training accuracy and 99% validation accuracy. CNN and L1 regularization model training accuracy is 97%, and model validation accuracy is 99.2%. The Y-axis represents training and validation accuracy,

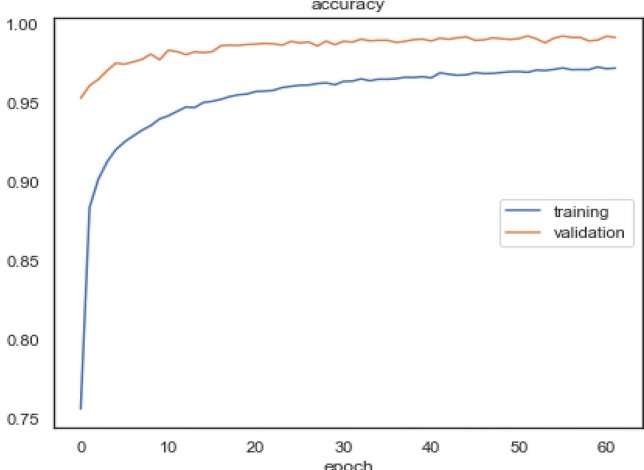

**Fig 2. Accuracy graph of 80−20% ratio of MNIST dataset.**

while the X-axis represents epoch count. It took 30 epochs for the model to converge on an optimal CNN and L1 regularization model for digit recognition. When we first started running our model, it gave 0.75% training and 0.47% validation accuracy, but as the epoch size increased over time, the results improved and improved until the model gave 97% training accuracy and 99% validation accuracy Fig 3 depicts the CNN and L1 regularization model's training and validation loss.

The CNN model has a training loss of 2.3 and a validation loss of 2. The Y-axis represents testing and validation loss, whereas the X-axis represents epoch count. The training loss of 2.3 and a validation loss of 2. Fig 4 shows the confusion matrix of the MNIST dataset.

The number of correct and incorrect predictions produced by a convolutional neural network and L1regularization. A confusion matrix is a table that is used to define how well a classification method performs. Column presents the predicted class and the actual class is presented in rows. The predicted class is represented in the column of the confusion matrix, whereas occurrences in the actual class are represented in the row. Numbers on the matrix diagonal indicate correct

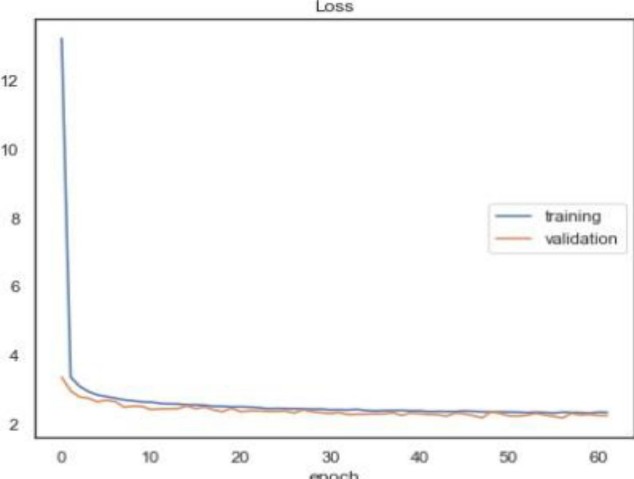

**Fig 3. Loss graph of 80−20% ratio of MNIST dataset.**

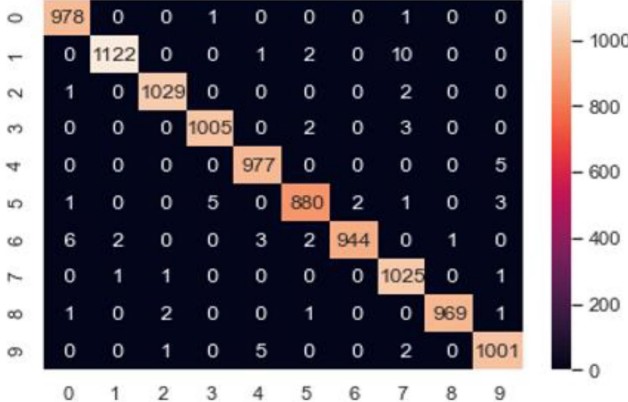

**Fig 4. Confusion matrix of 80−20% ratio of MNIST dataset.**

prediction, whereas values outside the matrix diagonal indicate incorrect prediction. The model has a training AUC value 99% and a validation value 100% in Fig 5.

The summarizes the CNN model's performance report, which is divided into 80% and 20% dataset testing and training ratios, with the training accuracy of the purposed model being 97.3% and the validation accuracy being 99.2%, respectively, with the sensitivity value for training 96.9 and the specificity value for training 99.8 and validation being 99.9. The accuracy factor for training is 97.8, whereas the precision factor for validation is 99.3. As a consequence, the recall value for training is 96.9, while the recall value for validation is 99.2.

On the MNIST dataset, the experiment is carried out using a **70−30%** ratio. The fluctuation model training accuracy is 96%, and model validation accuracy is 98% at 43 epochs, as shown in the graph. Fig 6. depicts the Y-axis representing training and validation accuracy and the X-axis. representing epoch count.

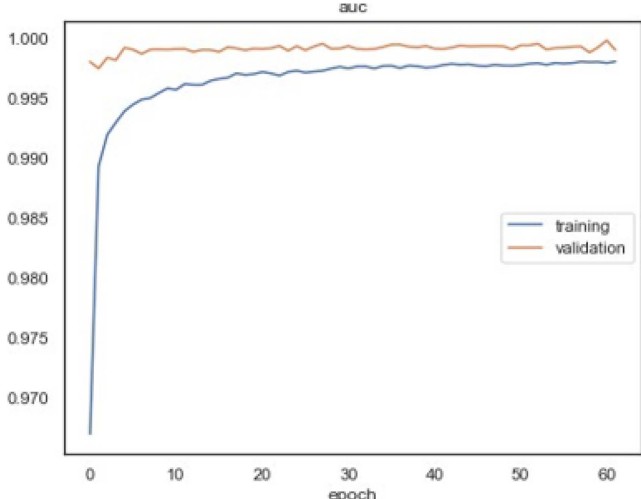

**Fig 5. AUC graph of 80−20% ratio of MNIST dataset.**

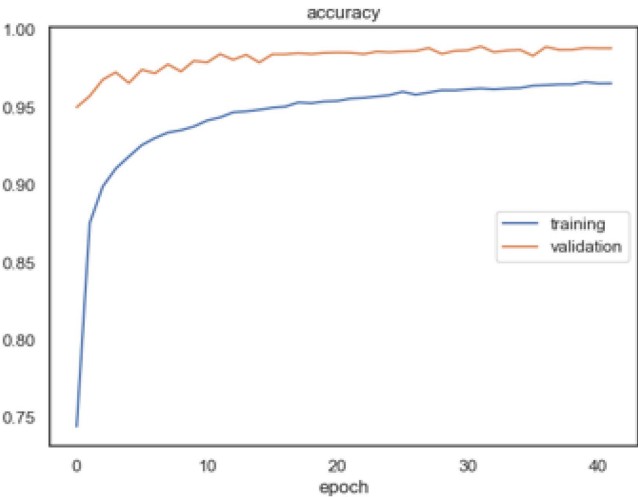

**Fig 6. Accuracy graph of 70−30 ratio of MNIST dataset.**

Fig 7 depicts the training and validation loss of the model. The training loss for the model is 2.3, and the validation loss is 2.2.

The model has a training AUC value 99.8% and a validation value 100% in Fig 8.

Fig 9 depicts the confusion matrix on the MNIST dataset. The number of correct and fault forecasts. A confusion matrix is a table that defines the performance of a classification algorithm.

The experiment is carried out using **90-10% ratio**. The graph depicts that the model training accuracy is 97.1% and the model validation accuracy is 99.2% at 41 epochs. Fig 10 depicts the Y-axis representing training and validation accuracy and the X-axis representing epoch count.

Fig 11 depicts the model's training and validation loss. The model has a training loss of 2.3 and a validation loss of 2.2. The Y-axis represents testing and validation loss, whereas the X-axis represents epoch count.

The model has a training AUC value of 99.8% and a validation value of 100% in Fig 12.

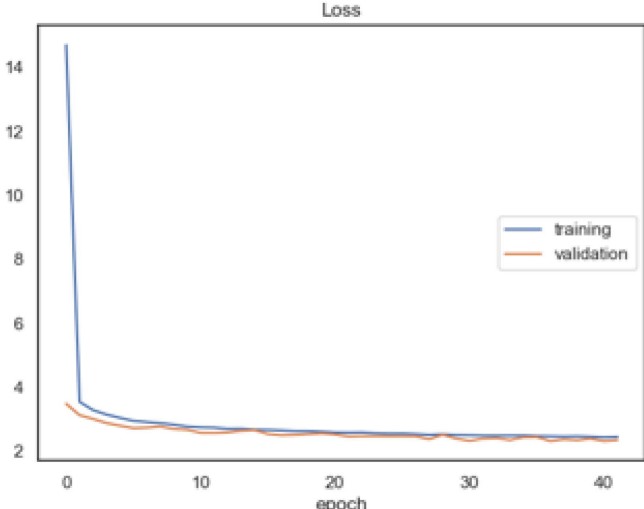

**Fig 7. Loss graph of 70−30 ratio of MNIST dataset.**

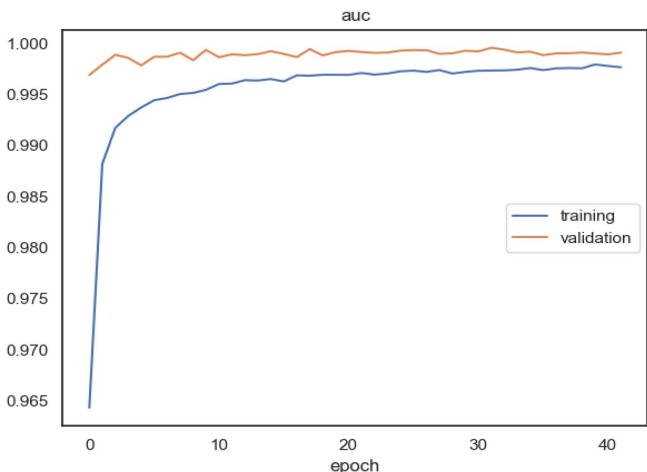

**Fig 8. AUC graph of 70−30 ratio of MNIST dataset.**

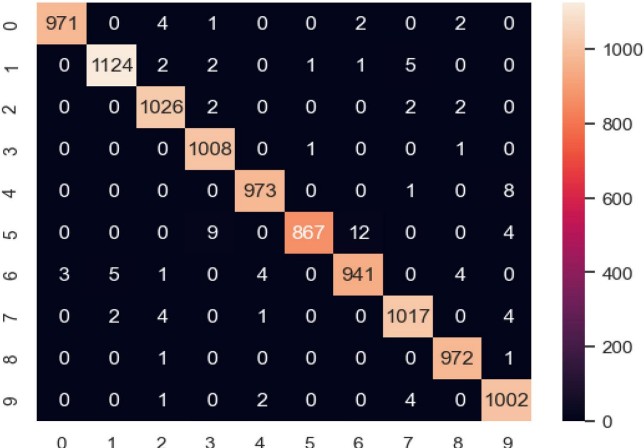

**Fig 9. Confusion matrix of 70−30 ratio of MNIST dataset.**

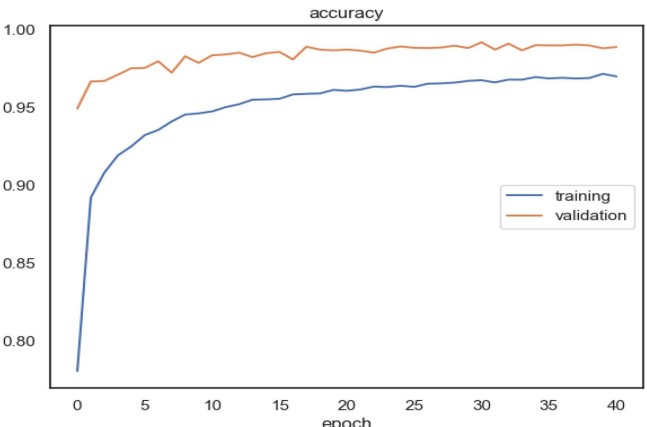

**Fig 10. Accuracy graph of 90−10 ratio of MNIST dataset.**

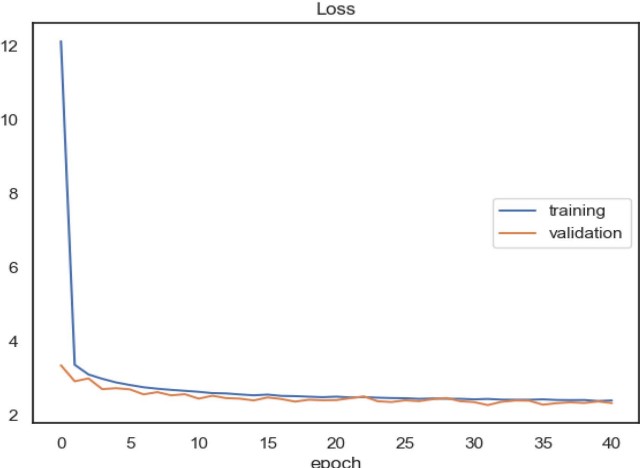

**Fig 11. Loss graph of 90−10 ratio of MNIST dataset.**

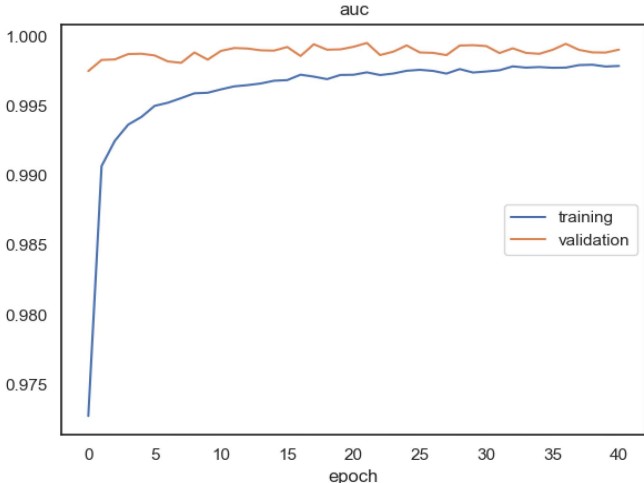

**Fig 12. AUC graph of 90−10 ratio of MNIST dataset.**

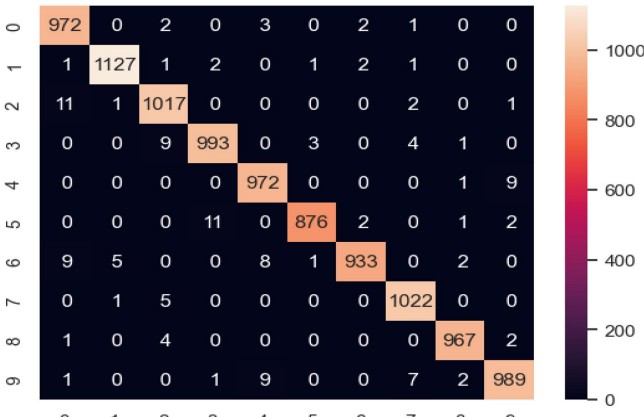

**Fig 13. Confusion matrix of 90−10 ratio of MNIST dataset.**

Fig 13 depicts the confusion matrix on the MNIST dataset.

The experiment is carried out using **60−40% ratio**. The graph depicts that the model training accuracy is 96% and the model validation accuracy is 98%. Fig 14 depicts the Y-axis representing training and validation accuracy and the X-axis representing epoch count.

Fig 15 depicts the model's training and validation loss. The model has a training loss of 2.3 and a validation loss of 2.1. The Y-axis represents testing and validation loss, whereas the X-axis represents epoch count. The loss values varies with the learning rate. If the pace of learning is sluggish, the loss value decreases gradually. If the learning rate is high, the loss value falls fast.

The model has a training AUC in Fig 16 value 99.8% and a validation value 99.9%.

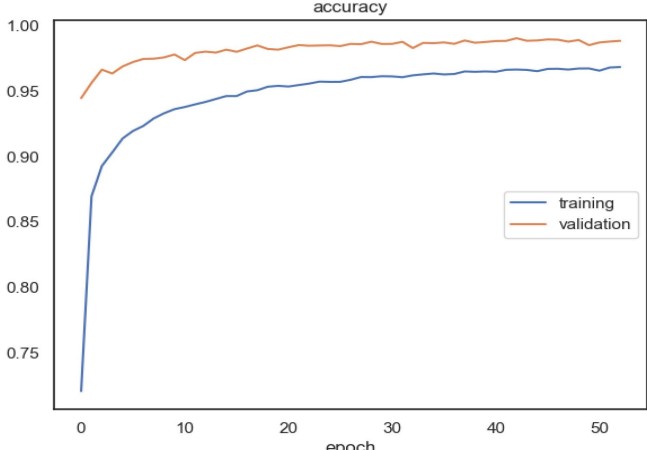

**Fig 14. Accuracy graph of 60−40 ratio of MNIST dataset.**

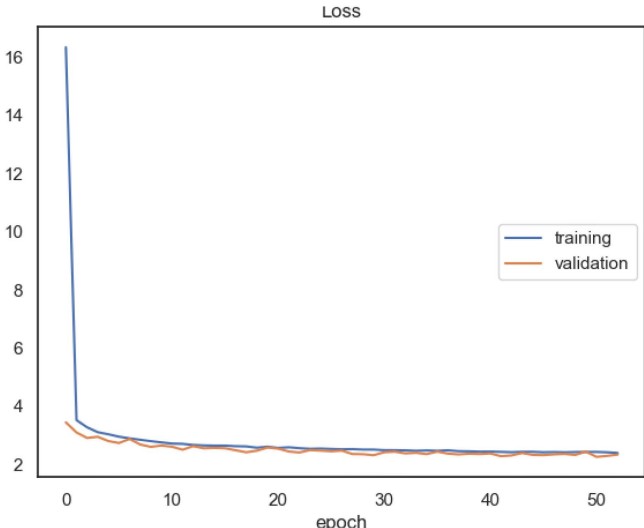

**Fig 15. Loss Graph of 60−40 ratio of MNIST dataset.**

The Fig 17 depicts the confusion matrix on the MNIST dataset.

The experiment is carried out using **50-50% ratio**. The Fig 18 depicts that the model training accuracy is 96.5% and the model validation accuracy is 98.8%. Fig 18 depicts the Y-axis representing training and validation accuracy and the X-axis representing epoch count.

Fig 19 depicts the model's training and validation loss. The model has a training loss of 2.3 and a validation loss of 2.1.

The model has a training AUC in Fig 20 value 99.8% and a validation value 99.9%.

The Fig 21 depicts the confusion matrix on the MNIST dataset.

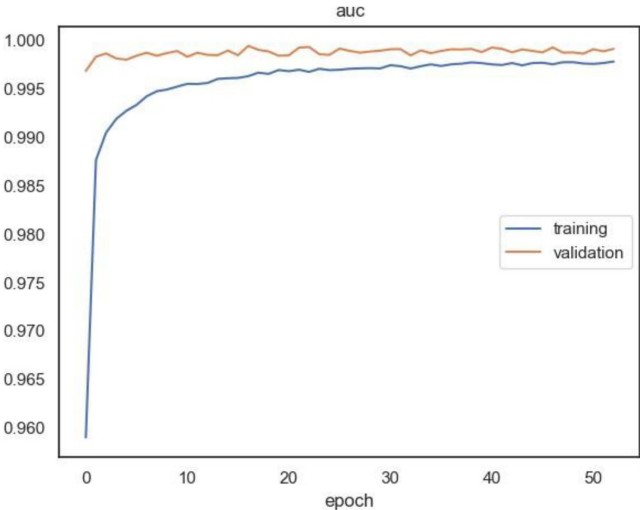

**Fig 16. AUC graph of 60−40 ratio of MNIST dataset.**

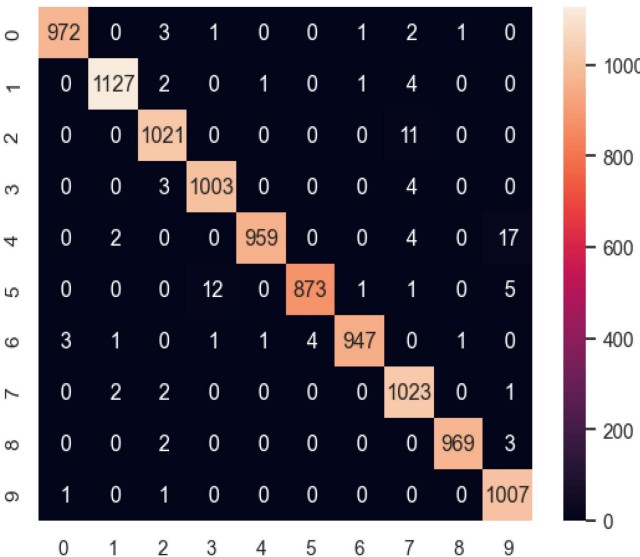

**Fig 17. Confusion matrix of 60−40 ratio of MNIST dataset.**

## Mango leaves images dataset

When applying L1 regularization with a coefficient value of 0.01 to the dense (fully connected) layers and a coefficient of 0.001 to the convolutional layers of a Convolutional Neural Network (CNN) trained on the Mango Tree Leaves dataset, the model undergoes a regularization process that encourages sparsity in both the convolutional and dense layers. In the context of the Mango Tree Leaves dataset, which likely contains images of mango tree leaves for classification or analysis, this dual L1 regularization strategy promotes feature selection in the convolutional layers, helping the model focus on the most relevant visual patterns in the leaves. Simultaneously, it encourages sparsity in the dense layers, reducing the

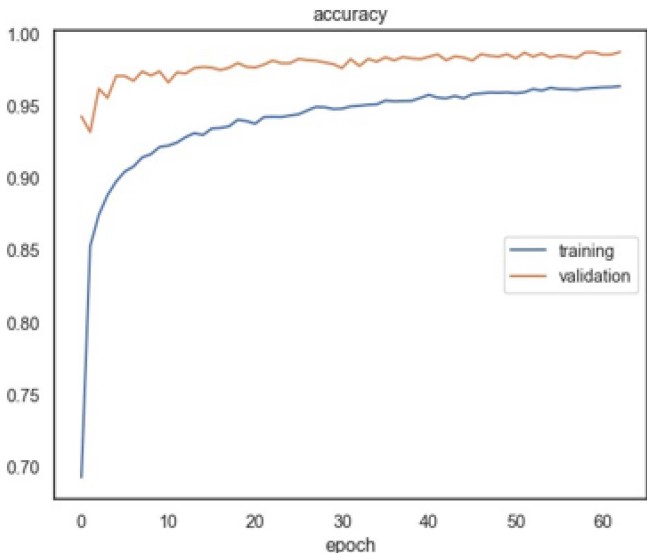

**Fig 18. Accuracy Graph of 50−50 ratio of MNIST dataset.**

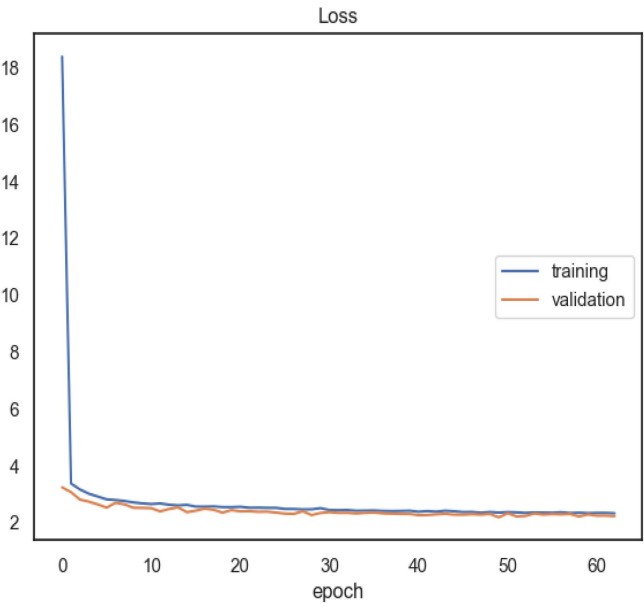

**Fig 19. Loss Graph of 50−50 ratio of MNIST dataset.**

complexity of the network's decision-making process. This regularization approach with different strength values in the convolutional and dense layers can potentially enhance the model's ability to generalize from the dataset, improve inter-pretability, and mitigate overfitting, ultimately aiding in more accurate classification or analysis of mango tree leaves. To get the best results on the mango leaf dataset, the convolutional neural network with L1 regularization has to be trained for about 9.8 hours (35,280 seconds) over 59 epochs.

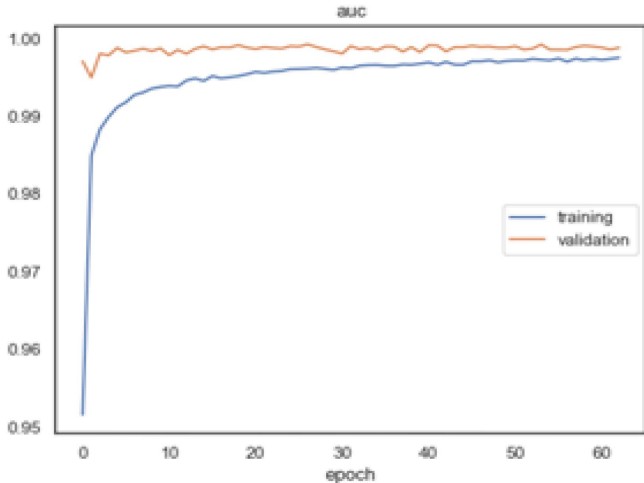

**Fig 20. AUC Graph of 50−50 ratio of MNIST dataset.**

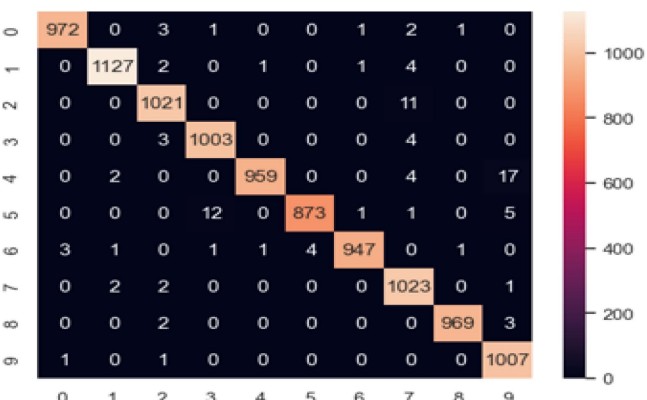

**Fig 21. Confusion matrix of 50−50 ratio of MNIST dataset.**

The experiment is performed for mango tree dataset. The experiment is performed for an **50−50% ratio** 23250 images belonging to 16 classes. The graph shows the fluctuation of training and validation accuracy of the CNN and L1 regularization model is 97%, shown in Fig 22.

CNN and L1 regularization model training accuracy is 92%, and model validation accuracy is 97% at 50 epochs. The training loss is 0.3, while the validation loss. is 0.1 shown in Fig 23.

The model has a training AUC value 99.7% and a validation value 100% shown in Fig 24.

The second experiment is performed for **60−40% ratio**. The graph shows the fluctuation model training accuracy is 92.6%, and model validation accuracy is 96%. The Y-axis represents training and validation accuracy, while the X-axis represents epoch count, Shown in Fig 25.

The training loss is 0.2, while the validation loss is 0.1 shown in Fig 26.

The model has a training AUC value 99.7% and a validation value 100% shown in Fig 27.

The model has a training AUC value 99.7% and a validation value 100% shown in Fig 28.

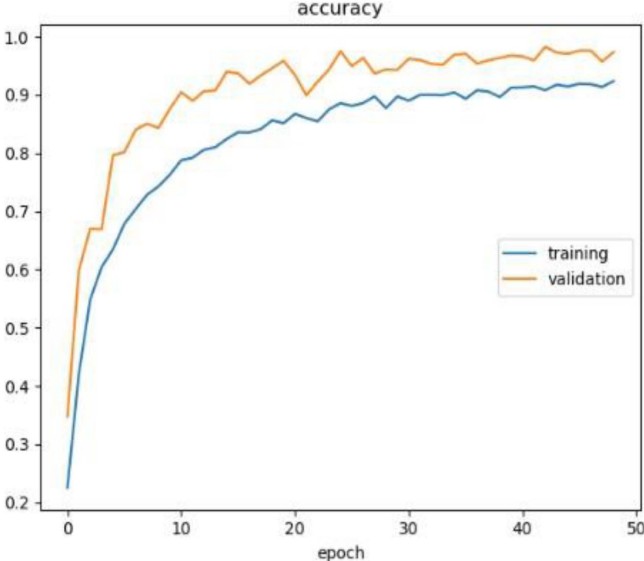

**Fig 22. Accuracy Graph 50−50 ratio of Mango leaves images dataset.**

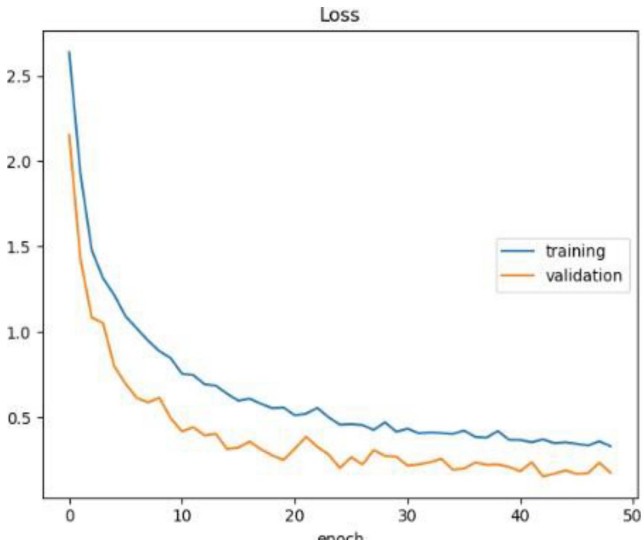

**Fig 23. Loss graph 50−50 ratio of Mango leaves images dataset.**

The experiment is performed for **70-30% ratio**. The graph shows the fluctuation model training accuracy is 93%, and model validation accuracy is 97%. shown in Fig 29.

The training and validation loss of the model are depicted in the figure. The training loss is 0.3, while the validation loss is 0.1 shown in Fig 30.

The model has a training AUC value 99.7% and a validation value 99.9% shown in Fig 31.

The experiment is performed for **80−20% ratio**. The graph shows the fluctuation model training accuracy is 93.4%, and model validation accuracy is 97% shown in Fig 32.

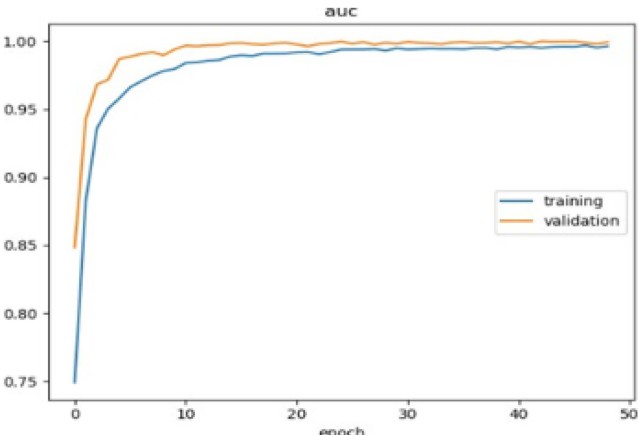

**Fig 24. AUC Graph 50−50 ratio of Mango leaves images dataset.**

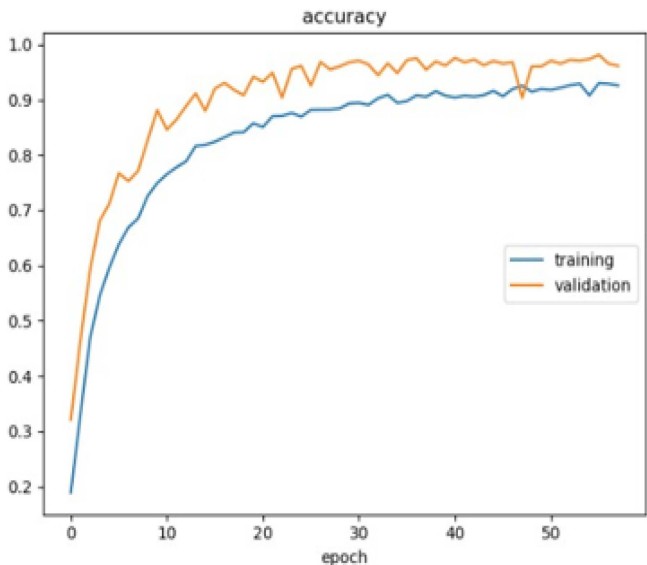

**Fig 25. Accuracy Graph 60−40 ratio of Mango leaves images dataset.**

The training and validation loss of the model are depicted in the figure. The model has a 0.2 training loss and a 0.1 validation loss shown in Fig 33. The model has a training AUC value 99% and a validation value 99% shown in Fig 34

The experiment is performed for **90-10% ratio**. The graph shows the fluctuation model training accuracy is 92%, and model validation accuracy is 96% shown in Fig 35.

The training and validation loss of the model are depicted in the figure. The model has a 0.3 training loss and a 0.1 validation loss shown in Fig 36.

The model has a training AUC value 99.6% and a validation value 100% shown in Fig 37.

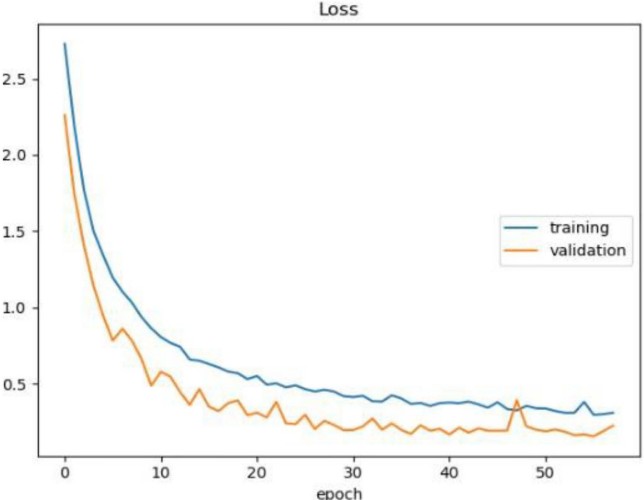

**Fig 26. Loss Graph 60−40 ratio of Mango leaves images dataset.**

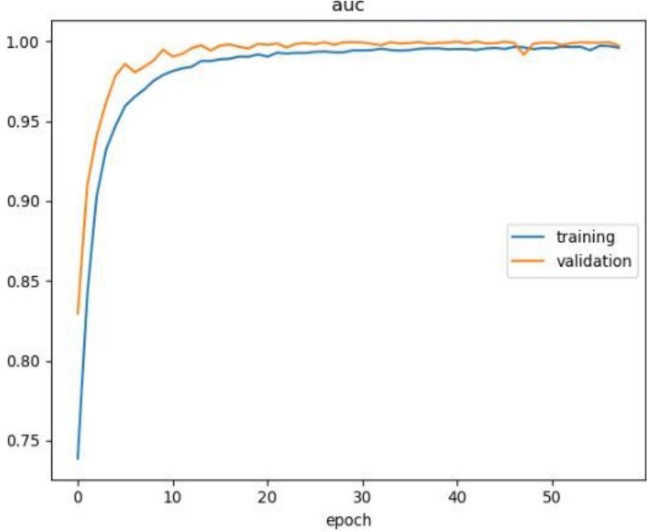

**Fig 27. AUC Graph 60−40 ratio of Mango leaves images dataset.**

## Hand-drawn sketches images dataset

The experiment is performed for hand-drawn sketches dataset. For training 16000 images belonging to 250 classes, and for validation 4000 images belonging to 250 classes utilized. When applying L1 regularization with a coefficient value of 0.001 to the dense (fully connected) layers of a Convolutional Neural Network (CNN) trained on hand-drawn sketches, such as the Quick, Draw! dataset, the model undergoes a process where the regularization term encourages many of the weight values in these dense layers to become small or even exactly zero. This promotes a form of feature selection, effectively simplifying the model's capacity to represent intricate details in the sketches. By selectively attenuating certain

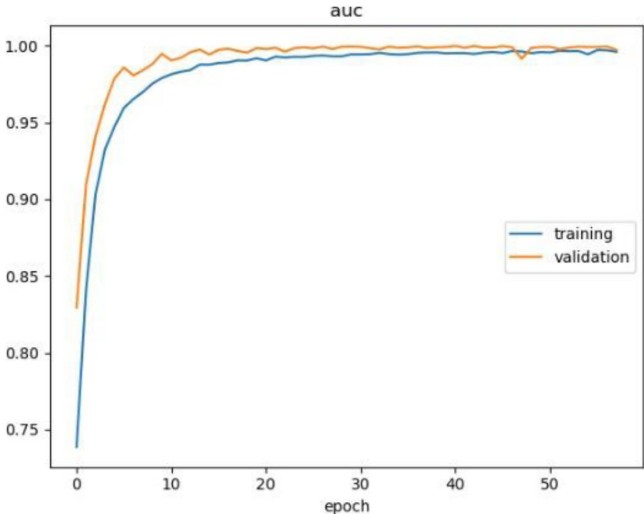

**Fig 28. AUC graph 60−40%.**

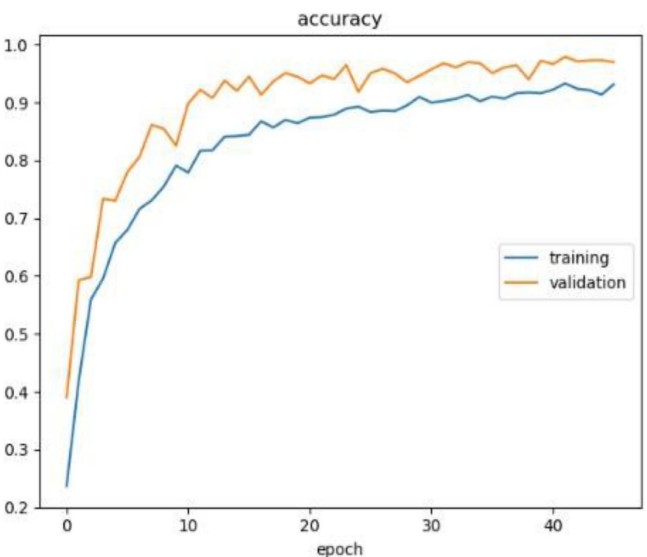

**Fig 29. Accuracy Graph 70−30 ratio of Mango leaves images dataset.**

connections, L1 regularization helps the CNN identify and emphasize the most relevant features while reducing the impact of less important ones. In the context of the Quick, Draw! dataset, which comprises millions of hand-drawn sketches across diverse categories, applying L1 regularization with a coefficient of 0.001 can lead to improved generalization and recognition accuracy by promoting a more concise and interpretable representation of the sketches. It took 123 seconds per epoch to train the CNN model with L1 regularization on the comic sketches dataset, for a total of around 2.05 hours (7380 seconds) for all 60 convergence epochs.

The experiment is performed for an **80−20% ratio.** The graph shows the fluctuation of training and validation accuracy of the CNN and L1 regularization model is 92.9%, as shown in Fig 38.

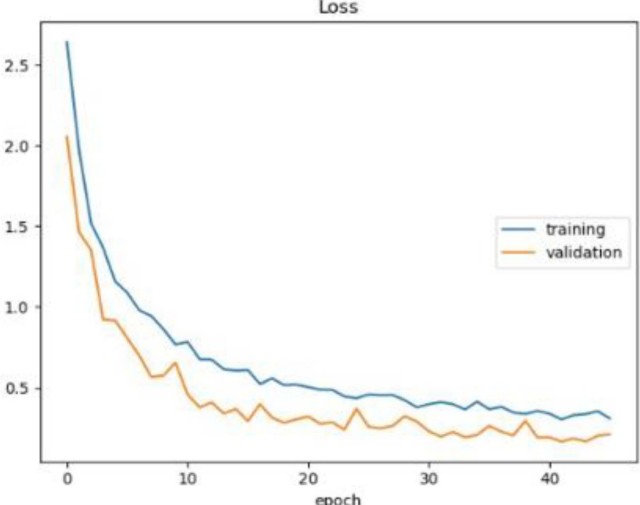

**Fig 30. Loss Graph 70−30 ratio of Mango leaves images dataset.**

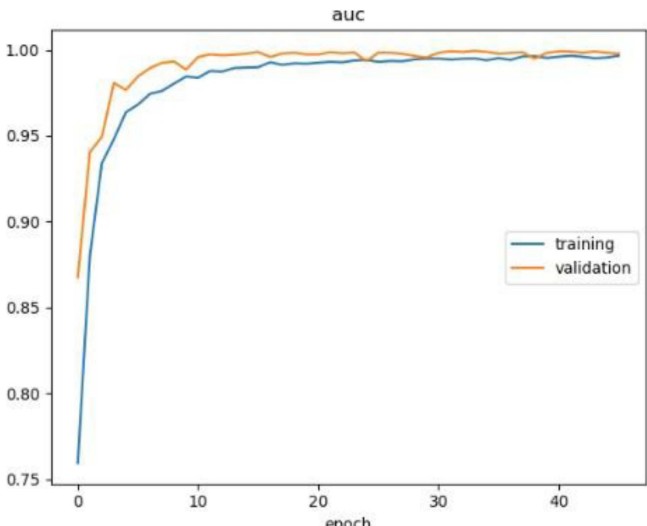

**Fig 31. AUC Graph 70−30 ratio of Mango leaves images dataset.**

CNN and L1 regularization model training accuracy is 92.5%, and model validation accuracy is 92.9%. Fig 39 depicts the CNN and L1 regularization model's training loss of 1.2% and validation loss 2%.

The Area Under the Curve (AUC) is a summary of the ROC curve that measures a classifier's ability to discriminate between classes. The greater the AUC, the better the model's ability to differentiate between positive and negative classifications. The training accuracy is 98% and the validation accuracy is 98% shown in Fig 40.

The experiment is performed for hand-drawn sketches dataset. For training 10000 images belonging to 250 classes for validation 10000 images belonging to 250 classes. The experiment is performed for a **50−50% ratio**. The graph shows the fluctuation of training and validation accuracy of the CNN and L1 regularization model is 92%, shown in Fig 41.

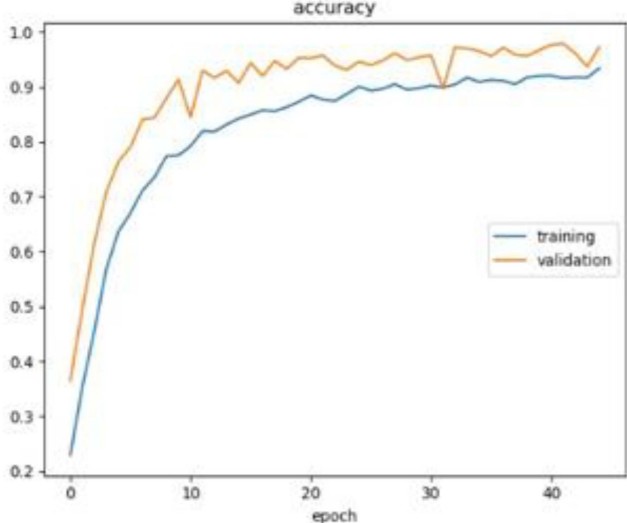

**Fig 32. Accuracy Graph 80−20 ratio of Mango leaves images dataset.**

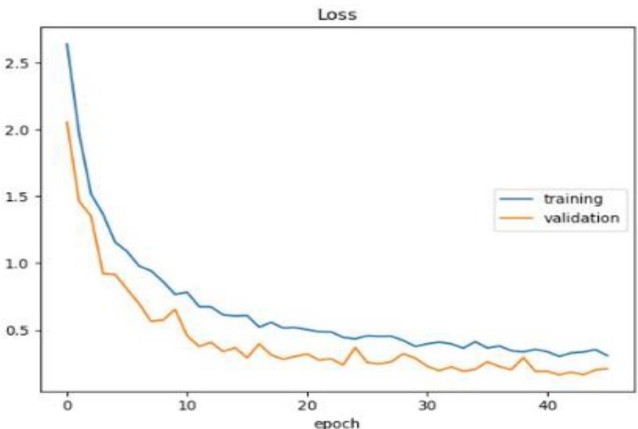

**Fig 33. Loss Graph 80−20 ratio of Mango leaves images dataset.**

CNN and L1 regularization model training accuracy is 91%, and model validation accuracy is 92%. Fig 42 depicts the CNN and L1 regularization model's training at 1.3% and validation loss at 1.3%.

Area Under the Curve (AUC) is a summary of the ROC curve that measures a classifier's ability to discriminate between classes. The greater the AUC, the better the model's ability to differentiate between positive and negative classifications. The training accuracy is 97% and the validation accuracy is 98% shown in Fig 43.

The experiment is performed for hand-drawn sketches dataset. For training 18000 images belonging to 250 classes for validation 2000 images belonging to 250 classes. The experiment is performed for a **90−10% ratio**. The graph shows the fluctuation of training and validation accuracy of the CNN and L1 regularization model is 92%, as shown in Fig 44.

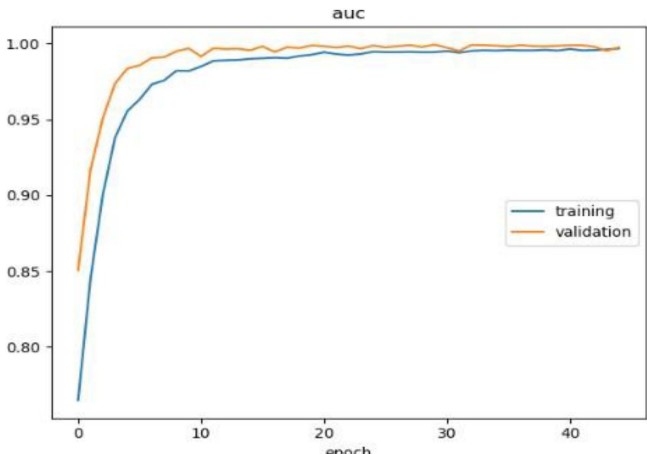

**Fig 34. AUC Graph 80−20 ratio of Mango leaves images dataset.**

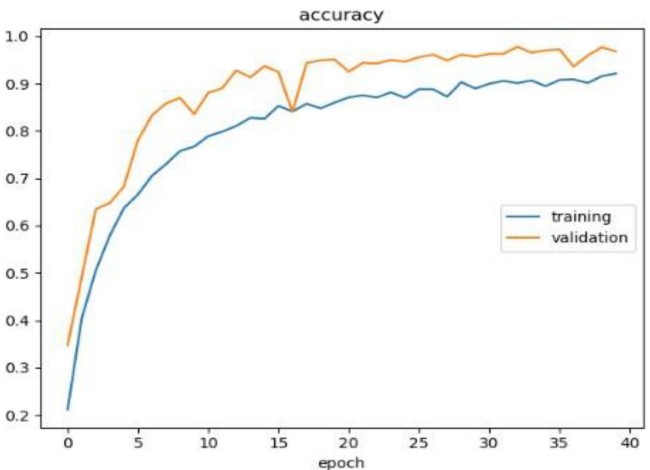

**Fig 35. Accuracy Graph 80−20 ratio of Mango leaves images dataset.**

CNN and L1 regularization model training accuracy is 91%, and model validation accuracy is 92%. The Y-axis represents training and validation accuracy, while the X-axis represents epoch count. Fig 45 depicts the CNN and L1 regularization model's training at 1.3% and validation loss at 1.2%.

The Area Under the Curve (AUC) is a summary of the ROC curve that measures a classifier's ability to discriminate between classes. The greater the AUC, the better the model distinguishes between positive and negative classifications. The validation AUC is 98% while the training AUC is 97% shown in Fig 46.

The experiment is performed for a **70-30% ratio**. The graph shows the fluctuation of training and validation accuracy of the CNN and L1 regularization model is 92%, shown in Fig 47.

CNN and L1 regularization model training accuracy is 92.6%, and model validation accuracy is 92.9%. The Y-axis represents training and validation accuracy, while the X-axis represents epoch count. Fig 48 depicts the CNN and L1 regularization model's training of 1.2% and validation loss of 1.2%.

The training accuracies are 99% and the validation accuracies are 98% shown in Fig 49.

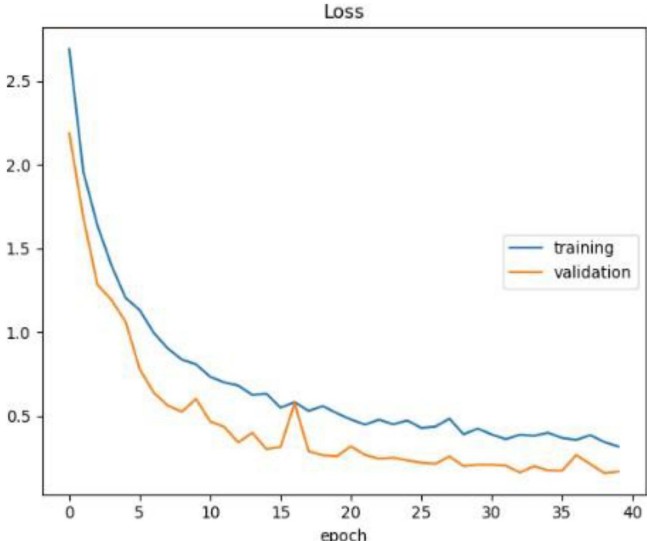

**Fig 36. Loss Graph 80−20 ratio of Mango leaves images dataset.**

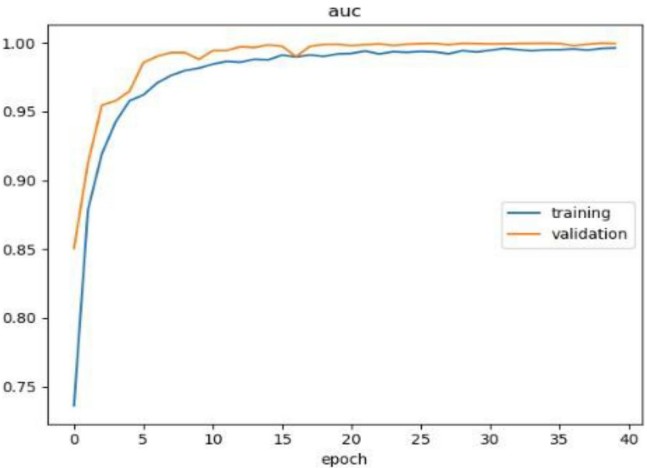

**Fig 37. AUC Graph 80−20 ratio of Mango leaves images dataset.**

## Other performance evaluation parameters

This section represents the Precision, Recall, Sensitivity, Specificity, and F1 Scores for MNIST, mango tree leave images, and hand-drawn sketch images 70−30%, 60−40%, 50−50%, and 80−20% ratios, respectively. The 80−20 split demonstrates slightly better performance in terms of validation accuracy compared to other splits because it provides an optimal balance between training and validation data. With 80% of data used for training, the model has sufficient samples to learn robust patterns while still retaining 20% for reliable validation. This split avoids the pitfalls of other ratios − 90−10 may have too little validation data for proper evaluation, while 50−50 provides insufficient training data. The improved accuracy with 80−20 suggests this ratio allows the model to better generalize to unseen data. While F1 scores remain similar across splits because they measure the balance between precision and recall (which stays relatively constant),

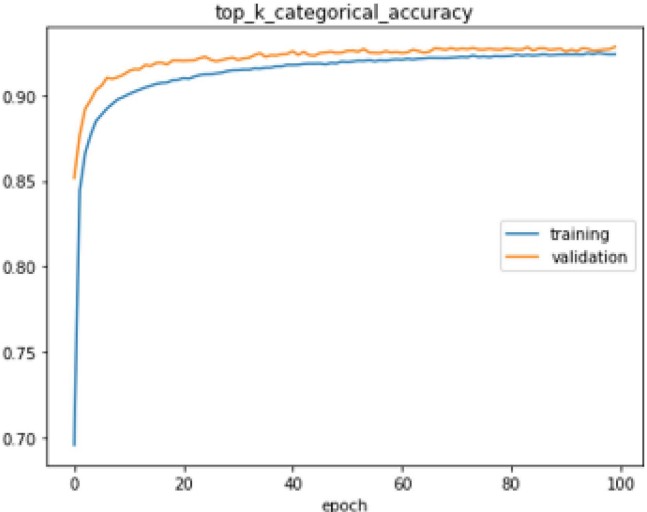

**Fig 38. Accuracy Graph 80−20 ratio of hand-drawn sketches dataset.**

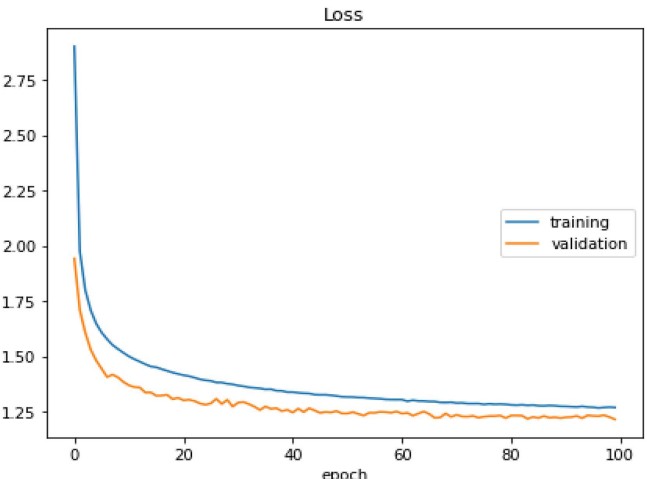

**Fig 39. Loss Graph 80−20 ratio of hand-drawn sketches dataset.**

accuracy benefits more noticeably from the larger training set in the 80−20 split. This indicates that while all splits perform adequately, the 80−20 ratio offers marginally superior learning conditions that translate to higher prediction correctness on validation data. The difference, though small, suggests 80−20 may be the most effective split when maximizing accuracy is the primary objective.

### MNIST dataset

First, we got the MNIST data set and divided it into four ratios for our experiment, i.e., 60:40, 50:50, and 80:20, and as it is seen, the best result we got is from the 80:20 ratio. Below, we should include all the results with their comparison.

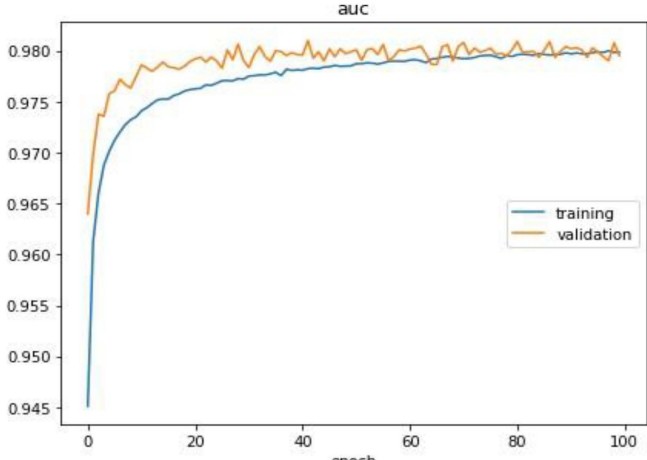

**Fig 40. AUC graph 80−20 ratio of hand-drawn sketches dataset.**

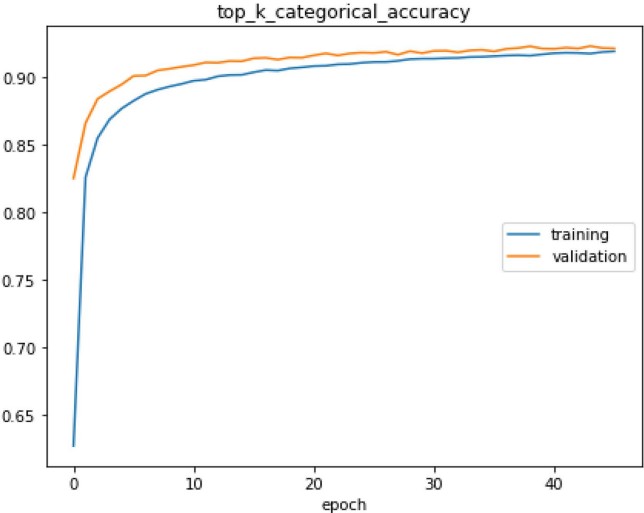

**Fig 41. Accuracy Graph 50−50 ratio of hand-drawn sketches dataset.**

The first bar shows the decision and its values are mentioned. The 80:20 ratio is represented blue bar, 70:30 is represented by the orange bar, the grey bar shows the 60:40 ratio, at last yellow represents 50:50. we used the same color for sensitivity, specificity, recall, and f1 score for their graphical representations. Scores of Performance Evaluation Parameters for MNIST dataset classification, with 70−30%; 60−40%; 50−50%, and 80−20% ratios are presented in Fig 50.

**Mango trees leave the dataset**

Second, we got the Mango tree leaves data set, and we divided it into four ratios for our experiment, i.e., 60:40, 50:50, 80:20, and as it is seen, the best results we got are from the 80:20 ratio. Below, we should include all the results with their comparison. The first bar shows the decision and its values are mentioned. The 80:20 ratio is represented blue bar, 70:30

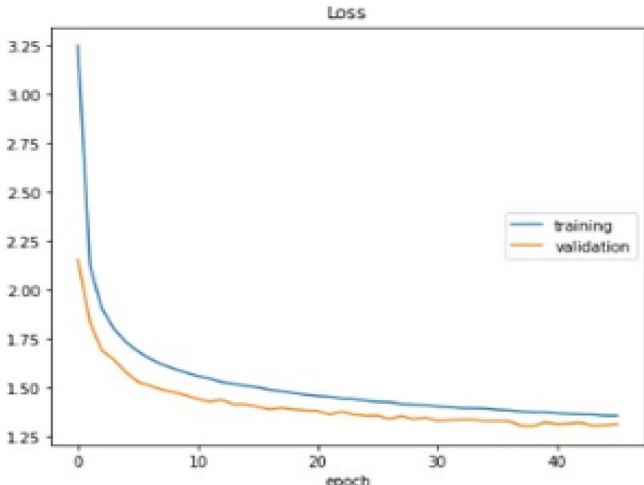

**Fig 42. Loss Graph 50−50 ratio of hand-drawn sketches dataset.**

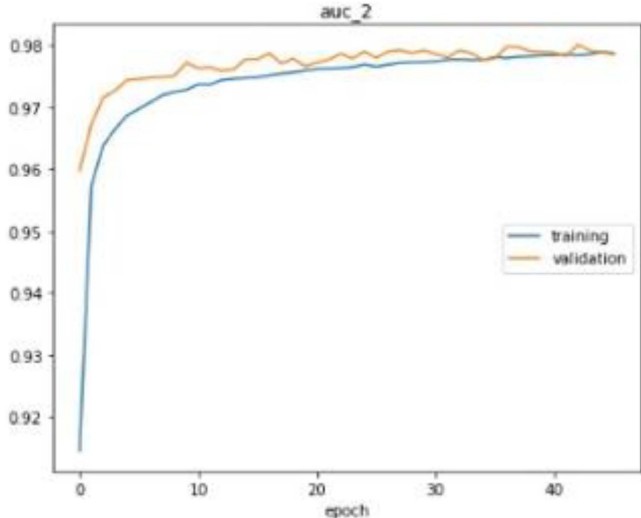

**Fig 43. AUC Graph 50−50 ratio of hand-drawn sketches dataset.**

is represented by the orange bar, and the grey bar shows the 60:40 ratio, at last yellow, represents 50:50. We used the same color for sensitivity, specificity, recall, and f1 score for their graphical representations. Scores of Performance Evaluation Parameters for mango tree leaves dataset classification, with 70−30%; 60−40%; 50−50%, and 80−20% ratios are presented in Fig 51.

### Hand-drawn sketches

Third, we got a Hand-drawn sketches set, and we divided it into four ratios for our experiment, i.e., 60:40, 50:50, 80:20, and as it is seen, the best results we got were from the 80:20 ratio. Below, we should include all the results with their

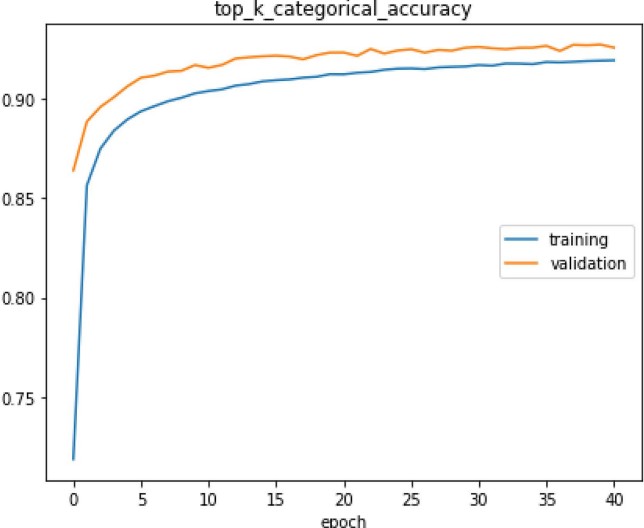

**Fig 44. Accuracy Graph 90−10 ratio of hand-drawn sketches dataset.**

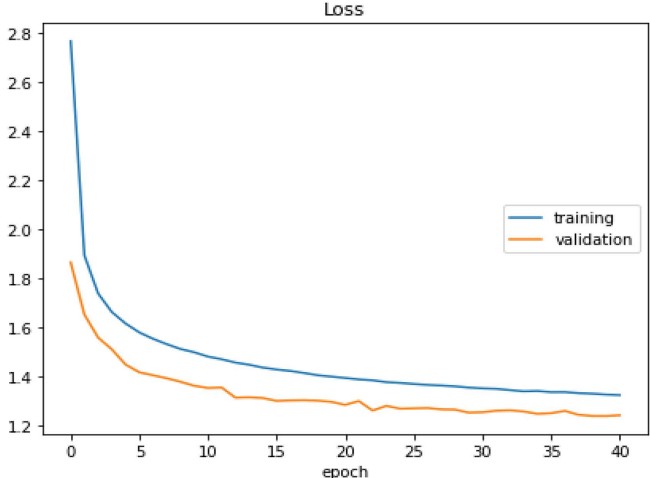

**Fig 45. Loss graph 90−10 ratio of hand-drawn sketches dataset.**

comparison. The first bar shows the decision and its values are mentioned. The 80:20 ratio is represented blue bar, 70:30 is represented by the orange bar, the grey bar shows the 60:40 ratio, at last yellow represents 50:50. We used the same color for sensitivity, specificity, recall, and f1 score for their graphical representations. Scores of Performance Evaluation Parameters for MNIST dataset classification, with 70−30%; 60−40%; 50–50%, and 80−20% ratios are presented in Fig 52.

a) Comparison

Within this segment, a comparison is drawn between the suggested model and previously employed deep learning and machine learning approaches. Specifically, our CNN with L1 regularization model targeting the MNIST dataset is

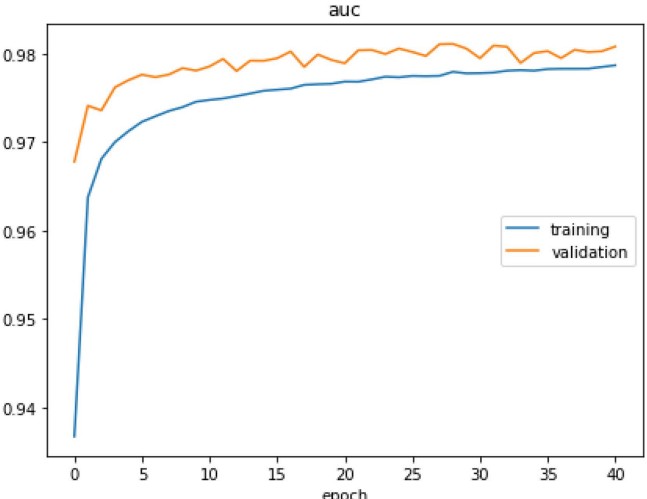

**Fig 46. AUC Graph 90−10 ratio of hand-drawn sketches dataset.**

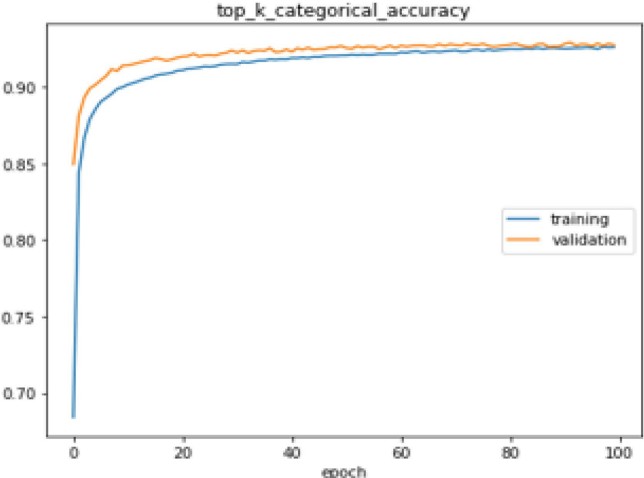

**Fig 47. Accuracy graph 70−30 ratio of hand-drawn sketches dataset.**

juxtaposed against K-nearest neighbors (KNN), Random Forest, and Convolutional Neural Network (CNN) models. The outcomes underscore the superiority of our model in terms of accuracy over these alternative deep learning and machine models, as visually depicted in Fig 53.

Hand-drawn sketches Images CNN with L1 regularization are compared with Convolutional neural network (CNN) and Alex-net for dataset. The accuracy of our model is more effective than other deep learning and machine models, as shown in Fig 54.

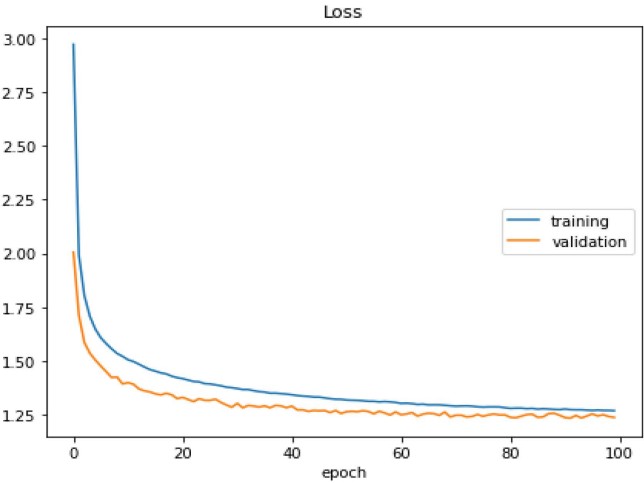

**Fig 48. Loss Graph 70−30 ratio of hand-drawn sketches dataset.**

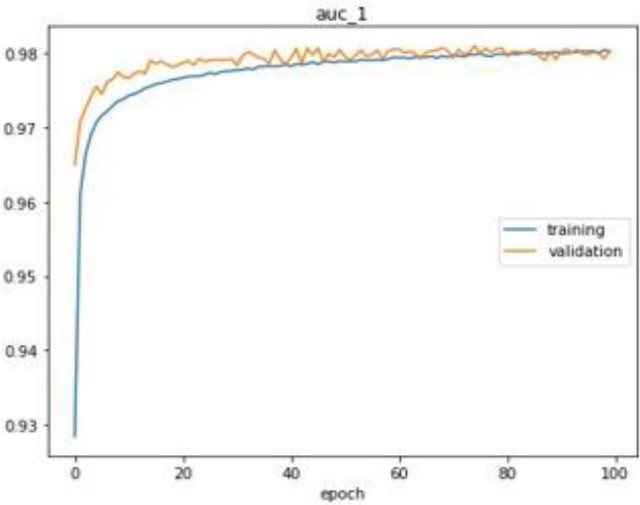

**Fig 49. AUC Graph 70−30 ratio of hand-drawn sketches dataset.**

For the dataset, Mango trees leave Images CNN with L1 regularization is compared with Convolutional neural network (CNN) Modified form VGG-16 and other CNN models. The accuracy of our model is more effective than other deep learning and machine models, as shown in Fig 55.

## Conclusion

A convolutional neural network (CNN) is a commonly used deep learning algorithm for image classification that excels in feature extraction and classifies objects based on those features. While other models are also hybrid with CNN, like SVM-CNN, LSTM-CNN still has overfitting challenges with large datasets. One major issue with CNNs is overfitting. To address this, we integrated L1 regularization into our CNN model and evaluated its performance over three datasets: (1) MNIST dataset (70,000 grayscale digits, split into 50K train, 10K validation, and 10K test), achieving a 99.9% training

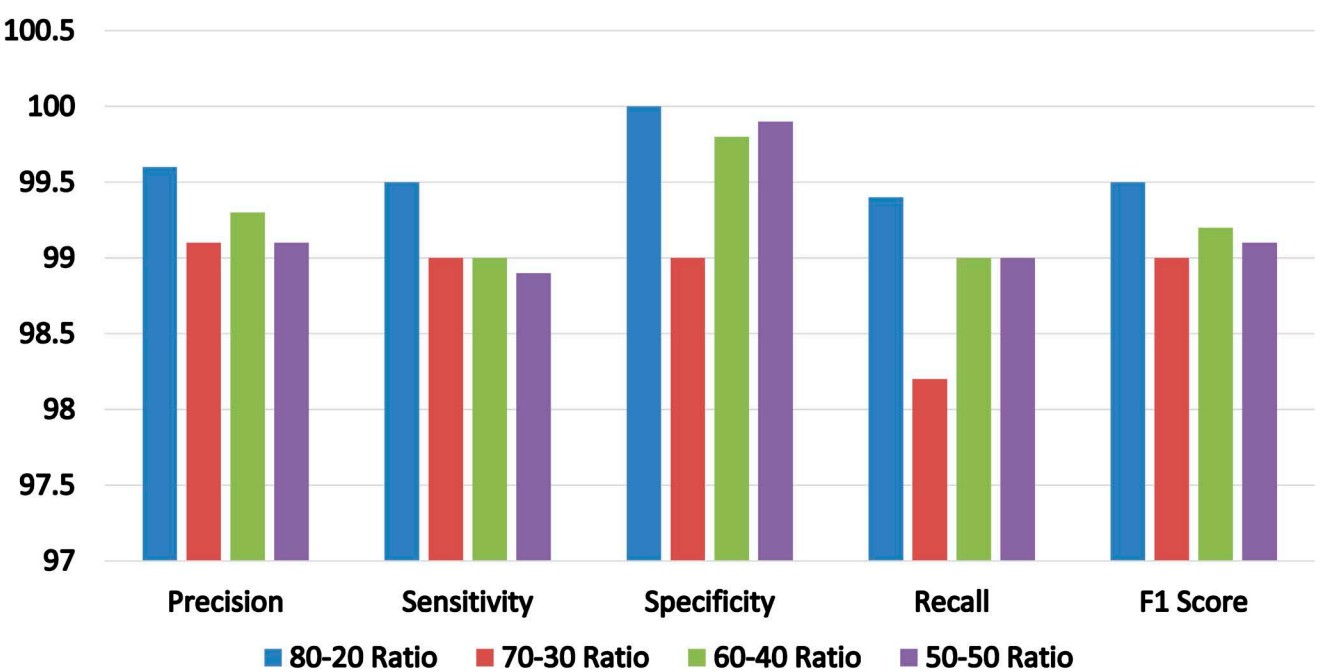

**Fig 50. Performances evolution score for MNIST dataset.**

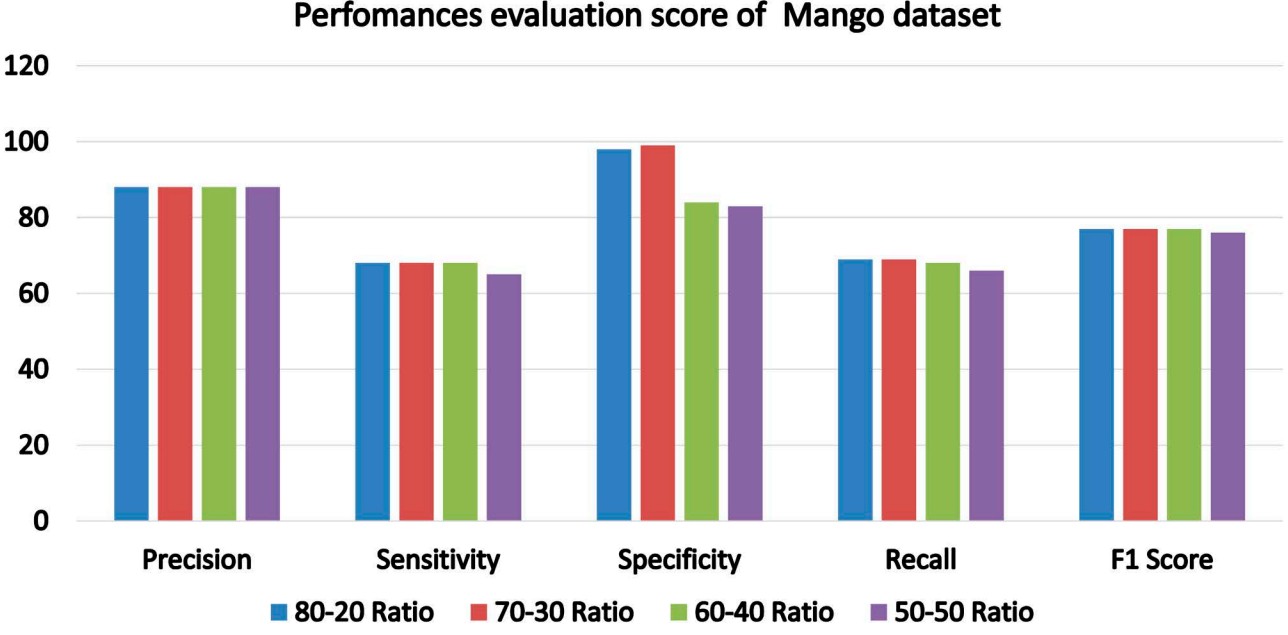

**Fig 51. Performances evolution score for Mango Leaves Images dataset.**

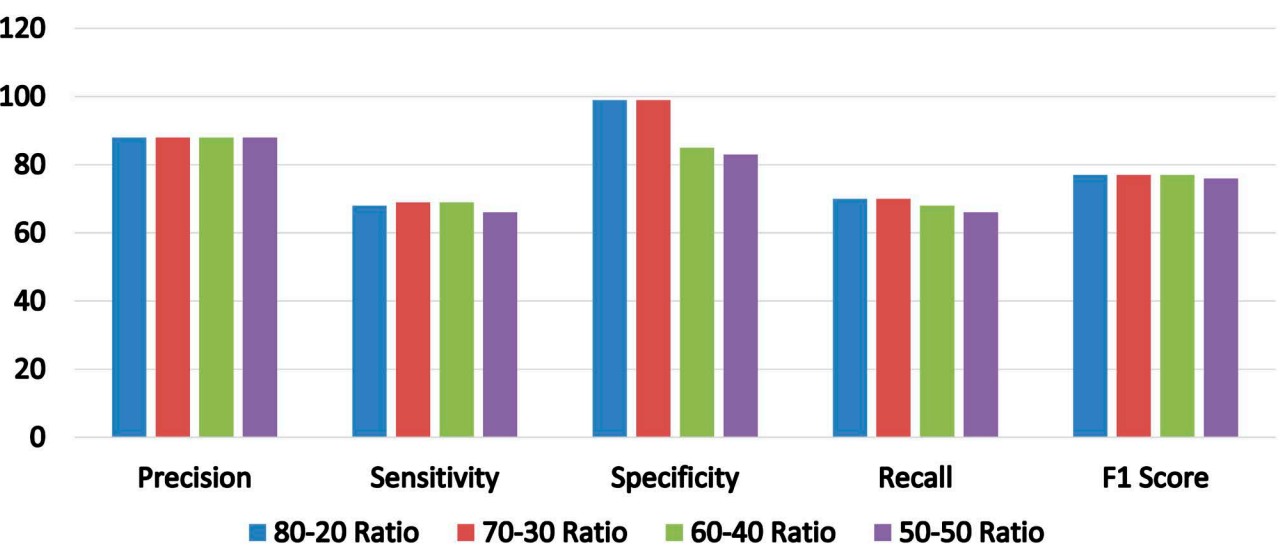

**Fig 52. Performances evolution score for Hand-Drawn Sketches images dataset.**

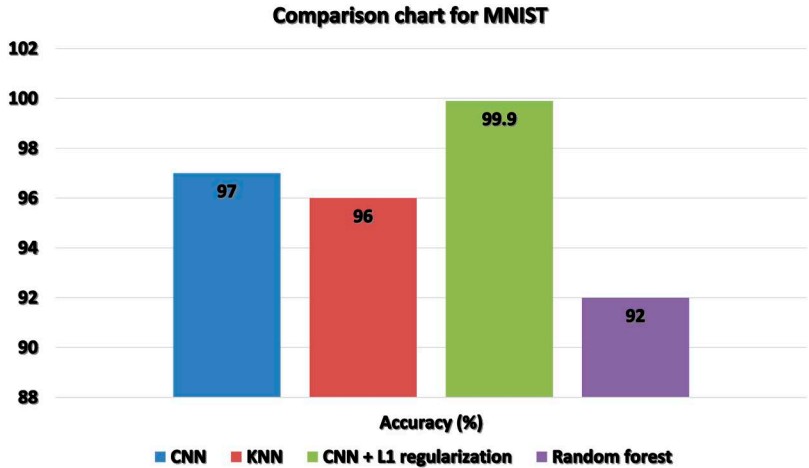

**Fig 53. Comparison chart for MNIST dataset.**

accuracy. (2) a mango leaf disease dataset (16 classes, 5000 images, augmented to balance classes), resulting in 97% accuracy. (3) Hand-drawn sketch images (20,000 images, preprocessed with edge detection) achieving 93% accuracy. In conclusion, the overall model performed well across the datasets and demonstrated improvements in accuracy by mitigating overfitting. In conclusion, our CNN model with L1 regularization performed well on all three datasets (MNIST, mango leaves, and hand-drawn drawings), dwindling overfitting and increasing accuracy. Because we utilized suitable data split ratios, conducted several tests, and evaluated against conventional models, the results are consistent. Our approach outperforms standard CNNs, as the data makes unambiguous. Further experiments with various datasets like VGG-16,

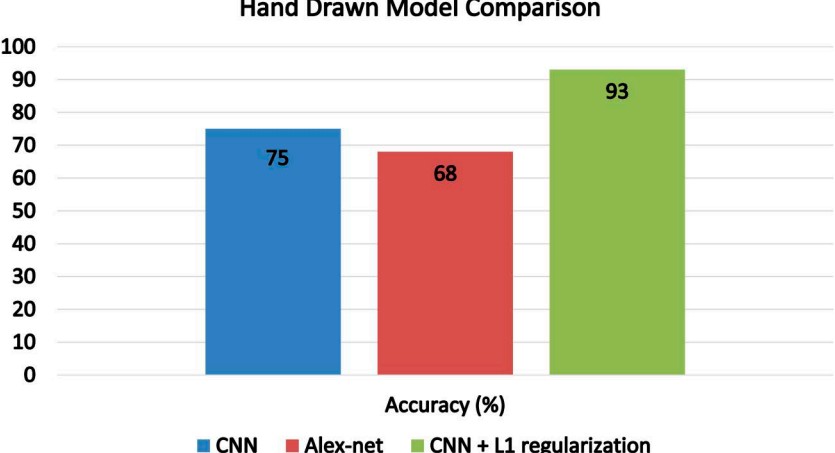

**Fig 54. Hand-drawn sketches Comparison chart.**

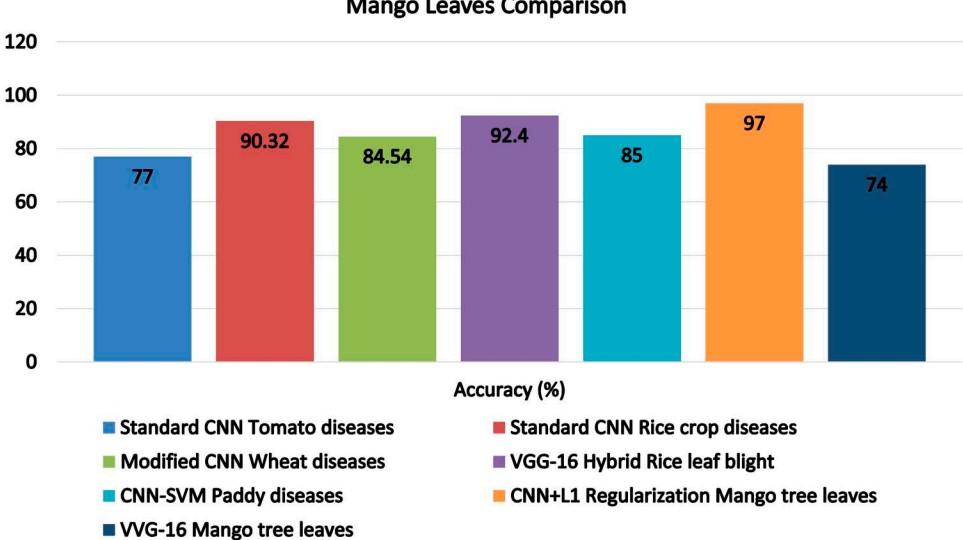

**Fig 55. Mango tree leaves comparison chart.**

Standard CNN, Modified CNN, and CNN-SVM would be advantageous; these results contribute to better image identification in agriculture and other domains. For future work, we will use a regularization hybrid approach.

## Author contributions

**Conceptualization:** Ramla Sheikh.

**Formal analysis:** Ahmed Alkhayyat.

**Funding acquisition:** Youngmoon Lee.

**Investigation:** Sikandar Ali, Youngmoon Lee.

**Methodology:** Ramla Sheikh, Fazli Wahid.

**Resources:** Sikandar Ali.

**Supervision:** Jawad Khan.

**Visualization:** Fazli Wahid, Yingling Ma.

**Writing – original draft:** Ramla Sheikh.

**Writing – review & editing:** Sikandar Ali, Jawad Khan.

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
