## [Decision Letter · Decision Letter 0]

3 Mar 2025

PONE-D-24-40423UNLOCKING THE POWER OF L1 REGULARIZATION: A NOVEL APPROACH TO TAMING OVERFITTING IN CNN FOR IMAGE CLASSIFICATIONPLOS ONE

Dear Dr. Khan,

Thank you for submitting your manuscript to PLOS ONE. After careful consideration, we feel that it has merit but does not fully meet PLOS ONE’s publication criteria as it currently stands. Therefore, we invite you to submit a revised version of the manuscript that addresses the points raised during the review process.

We look forward to receiving your revised manuscript.

Kind regards,

Jin Liu

Academic Editor

PLOS ONE

Journal Requirements:

“Hanyang University South Korea”

6. Please ensure that you refer to Figure 1, 33 and 34 in your text as, if accepted, production will need this reference to link the reader to the figure.

**Additional Editor Comments:**

Based on the advice received, this manuscript could be reconsidered for publication should the authors be prepared to incorporate suggestions in a major revision.

Reviewers' comments:

Reviewer's Responses to Questions

**Comments to the Author**

1. Is the manuscript technically sound, and do the data support the conclusions?

Reviewer #1: Partly

Reviewer #2: Partly

2. Has the statistical analysis been performed appropriately and rigorously? 

Reviewer #1: Yes

Reviewer #2: Yes

3. Have the authors made all data underlying the findings in their manuscript fully available?

Reviewer #1: Yes

Reviewer #2: Yes

4. Is the manuscript presented in an intelligible fashion and written in standard English?

Reviewer #1: No

Reviewer #2: Yes

5. Review Comments to the Author

Reviewer #1: 1. Introduction Lacks Professionalism and Clarity of Applications: The introduction is not written in a professional tone, and the applications mentioned in the article are unclear. While the title refers to image classification, it does not specify the type of images or datasets used. To improve clarity, explicitly state the purpose, dataset details, and real-world applications addressed by the study in the introduction.

2. Add a Literature Review Section and a Summary Table: Include a dedicated section for the literature review. Summarize key related works in a tabular format, highlighting aspects like the authors, methods, datasets used, strengths, and limitations. This will provide a structured and concise overview of existing research and help contextualize the contribution of the current study.

3. Proposed Method Lacks Clarity and Supporting Evidence: The description of the proposed method seems conceptual and lacks clarity regarding how different components are integrated. Provide detailed mathematical equations and explanations to support the methodology. This will ensure that readers can understand the logic and process behind the proposed approach.

4. The pseudo code 1 presented in the article needs additional explanation. Elaborate on each step of the algorithm, providing details about its purpose, functionality, and relevance to the proposed method. This will make the algorithm more accessible to readers.

5. The reference section does not include any articles from recent years. To reflect the state-of-the-art advancements, incorporate the latest research articles published in 2023, 2024, and 2025, particularly those relevant to the topic.

6. The references are poorly formatted, with inconsistencies in details like volume, issue, and page numbers. For instance, references 16 and 17 need correction. Ensure that all references adhere to a consistent and professional citation style, such as IEEE or APA, as per the article’s requirements.

7. The overall formatting of the paper is substandard. Revise the layout, including headings, subheadings, figures, tables, and spacing, to ensure consistency and adherence to the journal's formatting guidelines. Proper formatting enhances readability and professionalism.

Reviewer #2: 1.In the training on the MNIST dataset, the article mentions that it takes 60 epochs to converge to the optimal convolutional network. Given that MNIST is a large dataset, training the model could take a considerable amount of time. Could the authors provide the total training time as well as the average time the model takes to test each sample?

2.The article does not clearly explain the reasons and motivations for splitting the dataset into training and validation sets with different proportions, nor does it further analyze why the 80%-20% ratio produces the best results. Based on the results, the F1 scores for the four splitting methods are similar. If this part of the experiment holds any special significance, we would appreciate further clarification from the authors.

3.In the comparative experiments, the proposed CNN-L1 regularization method is compared with other different methods across three datasets, rather than comparing the same method across all three datasets. This weakens the persuasiveness of the experimental results, especially on the Mango trees leave image dataset, where it is only compared with a single VGG-16 model.

4.The article has issues such as missing chapter numbers and inconsistent reference formatting.

6. PLOS authors have the option to publish the peer review history of their article (what does this mean? ). If published, this will include your full peer review and any attached files.

**Do you want your identity to be public for this peer review?** For information about this choice, including consent withdrawal, please see our Privacy Policy .

Reviewer #1: No

Reviewer #2: No

---

## [Author Response · Author response to Decision Letter 1]

15 Apr 2025

Original Manuscript ID: PONE-D-24-40423

Original Article Title: “UNLOCKING THE POWER OF L1 REGULARIZATION: A NOVEL APPROACH TO TAMING OVERFITTING IN CNN FOR IMAGE CLASSIFICATION”

To: PLOS ONE Editor

Re: Response to reviewers

Dear Editor,

We sincerely thank you and the esteemed reviewers for your insightful comments and constructive suggestions. We have carefully revised the manuscript in accordance with all the recommendations provided, and we believe that the current version addresses the concerns raised and meets the required standards.

We are pleased to resubmit our revised manuscript for the original research article titled: “Unlocking the Power of L1 Regularization: A Novel Approach to Taming Overfitting in CNN for Image Classification.”

For your consideration, we are submitting the following documents:

(a) A detailed, point-by-point response to the reviewers’ comments (Response to Reviewers),

(b) A revised manuscript with changes highlighted in red (Supplementary Material for Review), and

(c) A clean version of the updated manuscript without highlights (Main Manuscript).

We hope that the revisions made will meet the approval of the editor and reviewers. Kindly find below a summary of the modifications and revisions incorporated into the manuscript.

Sincerely,

Khan et al.

Journal Requirements:

Author Response: Thank you for providing the exact PLOS ONE style template.

Author Action: We are working on the PLOS ONE style template, and will submit the revised paper in next round review.

Author Response: We revised Funding information in the manuscript, now it is aligned.

Author Action:

“Hanyang University South Korea”

Author Response: Professor Youngmoon Lee, one of the corresponding authors and the principal investigator of this research project, who is affiliated with Hanyang University South Korea, arranged the funding for this research project and played a significant role in writing and reviewing the manuscript. Beyond arranging the financial support and contributing to the manuscript, Hanyang University as the funding body had no direct role in the study design, data collection and analysis, or the decision to publish."

Author Action: This work was supported in part by the National Research Foundation of Korea (NRF) grant 2022R1G1A1003531, 2022R1A4A3018824 and Institute of Information and Communications Technology Planning and Evaluation (IITP) grant RS-2020-II201741, RS-2022-00155885, RS-2024-00423071 funded by the Korea government (MSIT). The authors would like to thank the Hanyang University, Republic of Korea Research for supporting this research work.

Author Response: Thank you for this clarification regarding the direct billing option. As the corresponding author Youngmoon Lee is affiliated with Hanyang University, which is directly related to billing, we meet the requirement for using this option.

Author Action: We have revised the funding statement and author affiliation in the manuscript accordingly.

Author Response: We used the following benchmark datasets, which are freely available online:

• MNIST: Available at https://www.kaggle.com/datasets/oddrationale/mnist-in-csv

• Mango Trees Leaf: Available at https://data.mendeley.com/datasets/94jf97jzc8/1

• Hand Drawn Sketches Images (Tree Category): Available at http://cybertron.cg.tuberlin.de/eitz/projects/classifysketch/

Paper code for replication will be available at https://github.com/hqsikandar/L1-REGULARIZATION-for-CNN.

Author Action: We included the above Data Availability statement in the revised manuscript accordingly.

6. Please ensure that you refer to Figure 1, 33 and 34 in your text as, if accepted, production will need this reference to link the reader to the figure.

Author response: Thank you for your insightful feedback.

Author action: We have referred to them in the text and double-checked everything.

Reviewers' comments:

1. Is the manuscript technically sound, and do the data support the conclusions?

Reviewer #1: Partly

Reviewer #2: Partly

Author Response: Thank you for this important guidance. We have carefully reviewed our manuscript to ensure it describes a technically sound piece of scientific research. We have incorporated all necessary details regarding the rigorous conduct of our experiments, including descriptions of appropriate controls, replication strategies, and the sample sizes used. Furthermore, we have meticulously reviewed the data presented to ensure that our conclusions are drawn appropriately and are fully supported by the evidence. We believe the revised manuscript now aligns with your requirements for technical soundness and data-supported conclusions."

Author Action: We have carefully reviewed manuscript accordingly. We revised conclusion and highlight with Red color, page 37

2. Has the statistical analysis been performed appropriately and rigorously?

Reviewer #1: Yes

Reviewer #2: Yes

Author response: Thank you for the compliments.

3. Have the authors made all data underlying the findings in their manuscript fully available?

Reviewer #1: Yes

Reviewer #2: Yes

Author response: Thank you for the compliments.

4. Is the manuscript presented in an intelligible fashion and written in standard English?

Reviewer #1: No

Reviewer #2: Yes

Author response: Thank you for the review.

Author action: This paper has been professionally reviewed by an English language professor.

Reviewer #1:

1. Introduction Lacks Professionalism and Clarity of Applications: The introduction is not written in a professional tone, and the applications mentioned in the article are unclear. While the title refers to image classification, it does not specify the type of images or datasets used. To improve clarity, explicitly state the purpose, dataset details, and real-world applications addressed by the study in the introduction.

Author Response: Thank you for your valuable feedback.

Author Action: We have carefully revised the introduction to enhance its professionalism, clarity, and specificity regarding applications, datasets, and research objectives. (Page 2,3)

2. Add a Literature Review Section and a Summary Table: Include a dedicated section for the literature review. Summarize key related works in a tabular format, highlighting aspects like the authors, methods, datasets used, strengths, and limitations. This will provide a structured and concise overview of existing research and help contextualize the contribution of the current study.

Author response: We sincerely appreciate your insightful recommendation to include a structured literature review and review table. This valuable suggestion has significantly strengthened our manuscript by providing a clearer context for our work within the existing research landscape.

Author action: The newly added sections directly address your feedback by: Literature Review heading and summary table heading, highlighted in red text for easy reference (Page 2,6)

3. Proposed Method Lacks Clarity and Supporting Evidence: The description of the proposed method seems conceptual and lacks clarity regarding how different components are integrated. Provide detailed mathematical equations and explanations to support the methodology. This will ensure that readers can understand the logic and process behind the proposed approach.

Author response: We sincerely appreciate your insightful feedback regarding the need for greater methodological clarity and mathematical consistency. Your comments have helped us significantly strengthen the technical presentation of our adaptive L1 regularization approach

Author action: we provide a detailed response addressing each of your concerns. We have expanded the mathematical foundation of our method to explicitly show how L1 regularization is applied, highlighted in red text for easy reference (Page 8).

4. The pseudo code 1 presented in the article needs additional explanation. Elaborate on each step of the algorithm, providing details about its purpose, functionality, and relevance to the proposed method. This will make the algorithm more accessible to readers.

Author response: Thank the reviewer's insightful suggestions regarding Pseudo Code 1, which have enabled us to significantly enhance its clarity and completeness.

Author action: We have thoroughly revised the pseudo code to include detailed explanations of each step's purpose, functionality, and relevance to our method while maintaining its concise format, highlighted in red text for easy reference (Page 9).

5. The reference section does not include any articles from recent years. To reflect the state-of-the-art advancements, incorporate the latest research articles published in 2023, 2024, and 2025, particularly those relevant to the topic.

Author response: Our sincere thanks to the reviewer for this constructive suggestion about incorporating recent research

Author action: Update our references with pertinent 2023-2025 publications that better position our work within the current state of the field, highlighted in red text for easy reference (Page 40-48)

6. The references are poorly formatted, with inconsistencies in details like volume, issue, and page numbers. For instance, references 16 and 17 need correction. Ensure that all references adhere to a consistent and professional citation style, such as IEEE or APA, as per the article’s requirements.

Author response: We sincerely appreciate the reviewer’s careful attention to detail regarding reference formatting.

Author action: We acknowledge the inconsistencies in our citations and have thoroughly revised the References section to ensure complete adherence to the IEEE style (as required by the journal). Highlighted in red text for easy reference (Page 40-48)

7. The overall formatting of the paper is substandard. Revise the layout, including headings, subheadings, figures, tables, and spacing, to ensure consistency and adherence to the journal's formatting guidelines. Proper formatting enhances readability and professionalism.

Author response: Thank you for the reviewer's thorough evaluation of our manuscript's formatting.

Author action: We recognize that proper presentation is essential for both readability and scholarly credibility. We revised the layout accorgingly.

Reviewer #2:

1. In the training on the MNIST dataset, the article mentions that it takes 60 epochs to converge to the optimal convolutional network. Given that MNIST is a large dataset, training the model could take a considerable amount of time. Could the authors provide the total training time as well as the average time the model takes to test each sample?

Author response: We sincerely appreciate the reviewer’s insightful question regarding computational efficiency.

Author action: In response, we have added the total training time and average test time per sample for all three datasets (MNIST, Mango Leaves, and QuickDraw) in the revised manuscript, highlighted in red text for easy reference. (Page 12, 21, 28).

2. The article does not clearly explain the reasons and motivations for splitting the dataset into training and validation sets with different proportions, nor does it further analyze why the 80%-20% ratio produces the best results. Based on the results, the F1 scores for the four splitting methods are similar. If this part of the experiment holds any special significance, we would appreciate further clarification from the authors.

Author response: We sincerely appreciate the reviewer’s insightful observation regarding our dataset splitting methodology.

Author action: In response, we have added clarifications in red text throughout the manuscript to better explain our experimental design choices and results. [Under heading Other performance evaluation parameters (Page 37).

3. In the comparative experiments, the proposed CNN-L1 regularization method is compared with other different methods across three datasets, rather than comparing the same method across all three datasets. This weakens the persuasiveness of the experimental results, especially on the Mango trees leave image dataset, where it is only compared with a single VGG-16 model.

Author response: Thank you for your review.

Author action: We have expanded our dataset comparison by including a mango tree dataset from agricultural fields along with other relevant datasets. This enhancement ensures that our model performs robustly across different farming environments. [figure 55].

4. The article has issues such as missing chapter numbers and inconsistent reference formatting, Location, book chapter

Au

---

## [Decision Letter · Decision Letter 1]

25 Jun 2025

UNLOCKING THE POWER OF L1 REGULARIZATION: A NOVEL APPROACH TO TAMING OVERFITTING IN CNN FOR IMAGE CLASSIFICATION

PONE-D-24-40423R1

Dear Dr. Khan,

We’re pleased to inform you that your manuscript has been judged scientifically suitable for publication and will be formally accepted for publication once it meets all outstanding technical requirements.

Kind regards,

Jin Liu

Academic Editor

PLOS ONE

Additional Editor Comments (optional):

After carring out the reviewers suggestions, this manuscript can be accepted for publication now.

Reviewers' comments:

Reviewer's Responses to Questions

**Comments to the Author**

1. If the authors have adequately addressed your comments raised in a previous round of review and you feel that this manuscript is now acceptable for publication, you may indicate that here to bypass the “Comments to the Author” section, enter your conflict of interest statement in the “Confidential to Editor” section, and submit your "Accept" recommendation.

Reviewer #1: All comments have been addressed

Reviewer #2: All comments have been addressed

2. Is the manuscript technically sound, and do the data support the conclusions?

Reviewer #1: Yes

Reviewer #2: Yes

3. Has the statistical analysis been performed appropriately and rigorously? 

Reviewer #1: Yes

Reviewer #2: Yes

4. Have the authors made all data underlying the findings in their manuscript fully available?

Reviewer #1: Yes

Reviewer #2: Yes

5. Is the manuscript presented in an intelligible fashion and written in standard English?

Reviewer #1: Yes

Reviewer #2: Yes

6. Review Comments to the Author

Reviewer #1: The author addressed all the comments and the manuscript has good research quality. I recommend the paper for publication.

Reviewer #2: The author has made revisions based on certain comments, and there is no other need for revision overall, so it is accepted.

7. PLOS authors have the option to publish the peer review history of their article (what does this mean? ). If published, this will include your full peer review and any attached files.

**Do you want your identity to be public for this peer review?** For information about this choice, including consent withdrawal, please see our Privacy Policy .

Reviewer #1: No

Reviewer #2: No
